# Rashomon Capacity: A Metric for Predictive Multiplicity in Classification

**Hsiang Hsu and Flavio P. Calmon**
John A. Paulson School of Engineering and Applied Sciences, Harvard University
hsianghsu@g.harard.edu, flavio@seas.harvard.edu

## Abstract

Predictive multiplicity occurs when classification models with statistically indistinguishable performances assign conflicting predictions to individual samples. When used for decision-making in applications of consequence (e.g., lending, education, criminal justice), models developed without regard for predictive multiplicity may result in unjustified and arbitrary decisions for specific individuals. We introduce a new metric, called Rashomon Capacity, to measure predictive multiplicity in probabilistic classification. Prior metrics for predictive multiplicity focus on classifiers that output thresholded (i.e., 0-1) predicted classes. In contrast, Rashomon Capacity applies to probabilistic classifiers, capturing more nuanced score variations for individual samples. We provide a rigorous derivation for Rashomon Capacity, argue its intuitive appeal, and demonstrate how to estimate it in practice. We show that Rashomon Capacity yields principled strategies for disclosing conflicting models to stakeholders. Our numerical experiments illustrate how Rashomon Capacity captures predictive multiplicity in various datasets and learning models, including neural networks. The tools introduced in this paper can help data scientists measure and report predictive multiplicity prior to model deployment.

## 1 Introduction

*Rashomon effect*, introduced by Breiman [1], describes the phenomenon where a multitude of distinct predictive models achieve similar training or test loss. Breiman reported observing the Rashomon effect in several model classes, including linear regression, decision trees, and small neural networks. In a foresighted experiment, Breiman noted that, when retraining a neural network 100 times on three-dimensional data with different random initializations, he "*found 32 distinct minima, each of which gave a different picture, and having about equal test set error*" [1, Section 8]. The set of almost-equally performing models for a given learning problem is called the *Rashomon set* [2, 3].

We focus on a facet of the Rashomon effect in classification problems called *predictive multiplicity*. Predictive multiplicity occurs when competing models in the Rashomon set assign conflicting predictions to individual samples [4]. Fig. 1 presents an updated version of Breiman's neural network experiment and illustrates predictive multiplicity in three classification tasks with different data domains and neural network architectures. Here, models that achieve statistically-indistinguishable performance on a test set assign wildly different predictions to an input sample. If predictive multiplicity is not accounted for, the output for this sample may ultimately depend on arbitrary choices made during training (e.g., parameter initialization).

Predictive multiplicity captures the potential individual-level harm introduced by an arbitrary choice of a single model in the Rashomon set. When such a model is used to support automated decision-making in sectors dominated by a few companies or Government—labeled *Algorithmic Leviathans* in [5, Section 3]—predictive multiplicity can lead to unjustified and systemic exclusion of individuals from critical opportunities. For example, an algorithm used for lending may deny a loan to a specific

36th Conference on Neural Information Processing Systems (NeurIPS 2022).

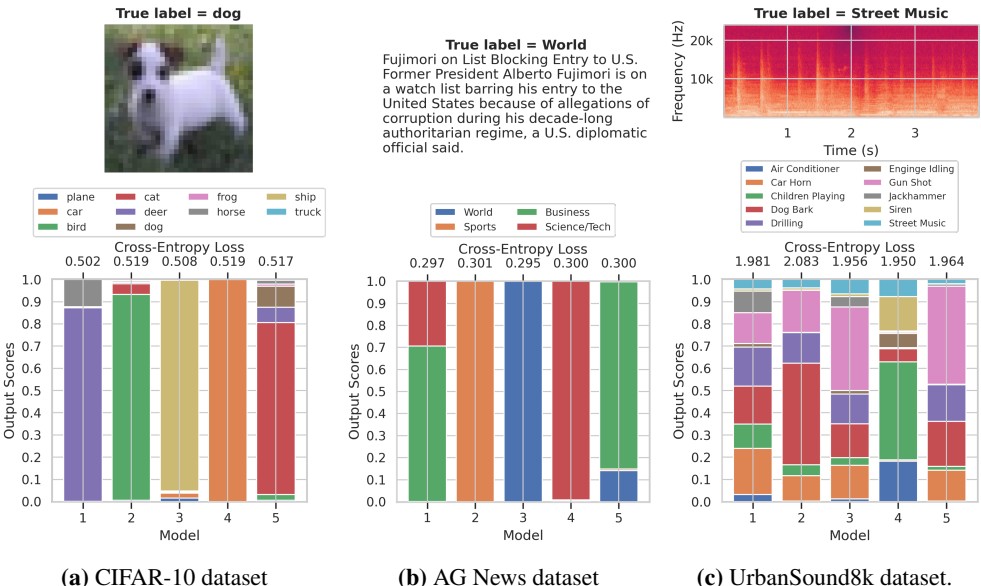

**Figure 1:** The scores (bottom) of a sample (top) generated by competing models. Predictive multiplicity occurs on different data domains and learning models, including an image dataset (CIFAR-10 [8]) trained with VGG16 [9], a natural language dataset (AG News [10]) trained with a simple neural networks after tokenization, and an audio dataset (UrbanSound8k [11]) trained with LSTM [12].

applicant. However, during model development, there may have been a competing model which performs equally well on average, yet would have approved the loan for this individual. As another example, Governments are increasingly turning to algorithms for grading exams that grant access to higher-level education (see, e.g., UK [6] and Brazil [7]). Here, again, accounting for predictive multiplicity is critical: an arbitrary choice of a single model in the Rashomon set may lead to an unwarranted restriction of educational opportunities to an individual student.

We introduce new methods for measuring and reporting predictive multiplicity in probabilistic classification. First, we postulate several properties that a predictive multiplicity metric must satisfy to simplify its interpretation by stakeholders. We then provide a new predictive multiplicity metric called *Rashomon Capacity*. Rashomon Capacity quantifies score variations among models in the Rashomon set for a given input sample. Unlike prior metrics restricted to thresholded scores (i.e., decisions), Rashomon Capacity can be applied to probabilistic classifiers that output a probability distribution over a set of classes (e.g., neural networks with soft-max output layers). Communicating such score disagreements helps stakeholders understand whether a given prediction is arbitrary, e.g., depending on randomness during training rather than patterns in the data.

We show that Rashomon Capacity of an input sample can be entirely captured by at most $c$ models in the Rashomon set, where $c$ is the number of predicted classes, *regardless of the size of the Rashomon set*. Remarkably, the computation of Rashomon Capacity also sheds light on a strategy for resolving predictive multiplicity. Instead of releasing a single model, we provide a greedy algorithm for identifying *a subset of models* in the Rashomon set that captures most of the score variations across a dataset. These models can be communicated to a stakeholder, empowering them to decide how to resolve conflicting scores via, e.g., randomization [5] and bagging [1]. In summary, our main contributions include:

1. We postulate desirable properties that *any* predictive multiplicity metric must satisfy. These properties motivate our definition of Rashomon Capacity and provide guidelines for the creation of new multiplicity metrics in future research. We also outline computational challenges in estimating predictive multiplicity in practice.

2. We introduce a new score-based metric for *quantifying* predictive multiplicity called Rashomon Capacity. Rashomon Capacity can be applied to measure score variations across competing classifiers that output either raw or thresholded (i.e. 0-1) scores.

3. We describe a methodology for *reporting* predictive multiplicity in probabilistic classification using Rashomon Capacity, with examples on different datasets and models. We advocate that predictive multiplicity must be reported to stakeholders in, for example, model cards [13].

4. We propose a procedure for *resolving* predictive multiplicity in probabilistic classifiers. Even though the Rashomon set may span a large (potentially uncountable) number of models, we show that the score variation for a sample is fully captured by a small subset of models in the Rashomon set. Communicating these predictions to stakeholders can empower them to decide how to resolve predictive multiplicity.

Omitted proofs, additional explanations and discussions, details on experiment setups and training, and additional experiments are included in Supplementary Materials (SM). Code to reproduce our experiments is available at https://github.com/HsiangHsu/rashomon-capacity.

## 2   Background and related work

**Notation.** We consider a dataset $\mathcal{D} = \{(\mathbf{x}_i, \mathbf{y}_i)\}_{i=1}^n$, e.g., a training or test set, for a classification task with $c$ classes/labels. Each sample pair $(\mathbf{x}_i, \mathbf{y}_i)$ is drawn i.i.d. from $P_{X,Y}$ with support $\mathcal{X} \times \Delta_c$. Here, $\Delta_c \triangleq \{(r_1, \cdots, r_c) \in [0,1]^c; \sum_{i=1}^c r_i = 1\}$ denotes the $c$-dimensional probability simplex. Let $\mathbf{e}_k$ be a length-$c$ indicator vector with one in the $k^{th}$ position and zero elsewhere, i.e., $[\mathbf{e}_k]_k = 1$, and $[\mathbf{e}_k]_j = 0 \ \forall j \neq k$, where $[\cdot]_j$ denotes the $j^{th}$ entry of a vector. Each $\mathbf{y}_i$ is one-hot encoded, i.e., $\mathbf{y}_i \in \{\mathbf{e}_k\}_{k=1}^c$. $\mathbb{1}(\cdot)$ denotes the indicator function.

We denote by $\mathcal{H}$ a hypothesis space, i.e., a set of candidate probabilistic classifier is parameterized by $\theta \in \Theta \subseteq \mathbb{R}^d$ that approximate $P_{Y|X=\mathbf{x}_i}$, i.e., $\mathcal{H} \triangleq \{h_\theta : \mathcal{X} \to \Delta_c : \theta \in \Theta\}$. The loss function used to evaluate model performance is denoted by $\ell : \Delta_c \times \Delta_c \to \mathbb{R}^+$ (e.g., cross-entropy) and $L(h_\theta) \triangleq \mathbb{E}_{P_{X,Y}}[\ell(h_\theta(X), Y)]$ the population risk. As usual, the population risk is approximated by the empirical risk $\hat{L}(h_\theta) \triangleq \frac{1}{n} \sum_{i=1}^n \ell(h_\theta(\mathbf{x}_i), \mathbf{y}_i)$.

**Rashomon set, Rashomon ratio, and pattern Rashomon ratio.** We define the Rashomon set as the set of all models in the hypothesis space that yield similar average loss. Formally, given a Rashomon parameter $\epsilon \geq 0$, the Rashomon set is defined as an $\epsilon$-level set [3]:

$$\mathcal{R}(\mathcal{H}, \epsilon) \triangleq \{h_\theta \in \mathcal{H}; L(h_\theta) \leq \epsilon\}. \tag{1}$$

Note that the Rashomon set is determined by the hypothesis space $\mathcal{H}$, the Rashomon parameter $\epsilon$, and also implicitly by the data distribution due to the evaluation of $L(h_\theta)$. The cardinality $|\mathcal{R}(\mathcal{H}, \epsilon)|$ or the volume[1] $\mathrm{vol}(\mathcal{R}(\mathcal{H}, \epsilon))$ of the Rashomon set (depending on whether the Rashomon set has finite elements) can be used to quantify the size of the Rashomon set. Given $\mathcal{R}(\mathcal{H}, \epsilon)$, the *Rashomon ratio* [3, Defn. 2] is defined as $\hat{\mathcal{R}}(\mathcal{H}, \epsilon) \triangleq \frac{\mathrm{vol}(\mathcal{R}(\mathcal{H},\epsilon))}{\mathrm{vol}(\mathcal{H})}$. $\hat{\mathcal{R}}(\mathcal{H}, \epsilon)$ represents the fraction of models in the hypothesis space that fit the data about equally well. A large Rashomon ratio indicates high multiplicity. Moreover, models with various desirable properties, such as better generalizability, can often exist inside a large Rashomon set. Similar to the Rashomon ratio, *pattern Rashomon ratio* [3, Defn. 12] is defined as the ratio of the count of all possible binary predicted classes given by the functions in the Rashomon set to that given by the functions in the hypothesis space. The complexity of computing pattern Rashomon ratio grows exponentially with the number of samples, yielding an "expensive" metric for predictive multiplicity when applied to large datasets.

**Ambiguity and discrepancy.** Instead of characterizing multiplicity in the hypothesis/ parameter space, Marx et al. [4] measure multiplicity in terms of the thresholded outputs (i.e., predicted classes) of a classifier, and propose two metrics: *ambiguity* and *discrepancy*. Ambiguity is the proportion of samples in a dataset that can be assigned conflicting predictions by competing classifiers in the Rashomon set. Discrepancy is the maximum number of predictions that could change in a dataset if we were to switch between models within the Rashomon set. More precisely, given a base model $\hat{h}$,

---

[1]Since $\mathcal{H}$ is parameterized, the volume of $\mathcal{R}(\mathcal{H}, \epsilon) = \{\theta \in \Theta; L(h_\theta) \leq \epsilon\}$ can be directly computed in $\mathbb{R}^d$.

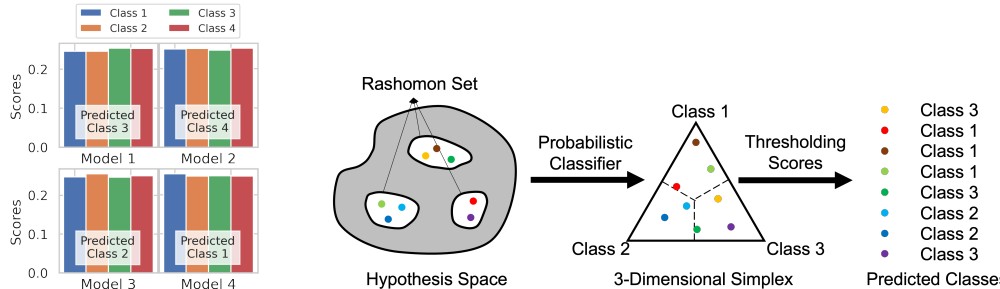

**Figure 2: Left**: 4 models output similar scores for 4 classes, but the predicted classes produced via thresholding are very different. This may lead to high ambiguity and/or discrepancy (cf. (2)). **Right**: Given a sample, the scores obtained from different models (colored dots) in the Rashomon set lead to different predicted classes by thresholding the scores (based on decision boundaries, i.e., dashed lines in the simplex). Prior work either measure multiplicity on the hypothesis set, e.g., the Rashomon ratio, or in terms of predicted classes, e.g., pattern Rashomon ratio and ambiguity/discrepancy. In this work, we measure predictive multiplicity directly in terms of scores on the probability simplex, i.e., either thresholded or raw scores.

the ambiguity $\alpha_\epsilon(\hat{h})$ and the discrepancy $\delta_\epsilon(\hat{h})$ are respectively defined as

$$\alpha_\epsilon(\hat{h}) \triangleq \frac{1}{n} \sum_{i=1}^{n} \max_{h \in \mathcal{R}(\mathcal{H}, \epsilon)} \mathbb{1}\left[\arg\max h(\mathbf{x}_i) \neq \arg\max \hat{h}(\mathbf{x}_i)\right],$$

$$\delta_\epsilon(\hat{h}) \triangleq \max_{h \in \mathcal{R}(\mathcal{H}, \epsilon)} \frac{1}{n} \sum_{i=1}^{n} \mathbb{1}\left[\arg\max h(\mathbf{x}_i) \neq \arg\max \hat{h}(\mathbf{x}_i)\right]. \tag{2}$$

For linear classifiers, both quantities in (2) can be estimated by mixed integer programming [4, Section 3]. A small $\epsilon$ could still lead to a large ambiguity, see Section SM. 2.2 for a discussion.

## 3 Measuring predictive multiplicity of probabilistic classifiers

The metrics in (2) for predictive multiplicity are based on predicted classes. Thus, they require finding the $\arg\max$ or thresholding the scores at the output of a classifier. In probabilistic classification, thresholding may mask similar predictions produced by competing models and artificially increase multiplicity: output scores can be almost equal across different classes, yet the (thresholded) predicted classes can be very different. For example, two scores $[0.49, 0.51]$ and $[0.51, 0.49]$ for a binary classification problem can lead to entirely different predicted classes after thresholding—1 and 0, respectively—and ultimately overestimate predictive multiplicity (see Fig. 2 (Left) for another multi-class example). In fact, predictive multiplicity metrics based on predicted classes may yield multiplicity *even for a single fixed model* when, for example, the threshold criteria for output scores is changed. This subtle, yet important difference motivates us to reconsider existing metrics and introduce a new predictive multiplicity metric that is applicable to both output scores and decisions (cf. Fig. 2 (Right) for an overview).

We begin this section by first outlining desirable properties of predictive multiplicity metrics for probabilistic classifiers. Motivated by the potential individual-level harm incurred by an arbitrary choice of model in the Rashomon set, we focus on per-sample multiplicity metrics. We formally define Rashomon Capacity in terms of the KL-divergence between the output scores of classifiers in the Rashomon set. We then use Rashomon Capacity to define a predictive multiplicity metric for individual samples in a dataset. We present these definitions in the ideal case where the population loss can be computed exactly, and discuss empirical approximation in Section 4.

### 3.1 Properties of multiplicity metrics for probabilistic classifiers

Consider a fixed data distribution $P_{X,Y}$ and a corresponding Rashomon set $\mathcal{R}(\mathcal{H}, \epsilon)$ for a classification problem with $c$ classes. For a given sample $\mathbf{x}_i \in \mathcal{D}$, we collect all possible output scores produced by models in $\mathcal{R}(\mathcal{H}, \epsilon)$, and define the $\epsilon$-*multiplicity set* as

$$\mathcal{M}_\epsilon(\mathbf{x}_i) \triangleq \{h(\mathbf{x}_i); h \in \mathcal{R}(\mathcal{H}, \epsilon)\} \subseteq \Delta_c. \tag{3}$$

Let $m(\cdot)$ be a measure of predictive multiplicity, and $m(\mathcal{M}_\epsilon(\mathbf{x}_i))$ be the predictive multiplicity of sample $\mathbf{x}_i$. Which properties should $m(\cdot)$ have? Ideally, we expect $m(\mathcal{M}_\epsilon(\mathbf{x}_i))$ to be a bounded value in $[1, c]$, since at least one class is assigned to sample $\mathbf{x}_i$, and at most $c$ different classes could be assigned to $\mathbf{x}_i$. Moreover, if $m(\mathcal{M}_\epsilon(\mathbf{x}_i)) = 1$ (i.e., a predictive multiplicity of 1), then one would expect that only one score is produced for $\mathbf{x}_i$ and, thus, all predictions in $\mathcal{M}_\epsilon(\mathbf{x}_i)$ are exactly the same. Similarly, if $m(\mathcal{M}_\epsilon(\mathbf{x}_i)) = c$ (i.e., predictive multiplicity equal to the number of classes), then there must exist $c$ models $\{h_1, \cdots, h_c\} \subseteq \mathcal{R}(\mathcal{H}, \epsilon)$ such that $h_j(\mathbf{x}_i) = \mathbf{e}_j$. In other words, each of the $c$ classes can be assigned to the sample, yielding a predictive multiplicity of $c$. Finally, $m(\mathcal{M}_\epsilon(\mathbf{x}_i))$ should be monotonic in $\mathcal{M}_\epsilon(\mathbf{x}_i)$, i.e., if $\mathcal{M}_\epsilon(\mathbf{x}_i) \subseteq \mathcal{M}'_\epsilon(\mathbf{x}_i)$, then $m(\mathcal{M}_\epsilon(\mathbf{x}_i)) \le m(\mathcal{M}'_\epsilon(\mathbf{x}_i))$. We summarize these desirable properties of predictive multiplicity metrics in the following definition.

**Definition 1.** *Let $\mathcal{P}_c$ be the power set[2] of the probability simplex $\Delta_c$. The function $m : \mathcal{P}_c \setminus \emptyset \to \mathbb{R}$ is a* predictive multiplicity metric[3] *if for any $\mathcal{A}, \mathcal{B} \in \mathcal{P}_c$*

1. *$1 \le m(\mathcal{A}) \le c$;*

2. *$m(\mathcal{A}) = 1$ if and only if $|\mathcal{A}| \le 1$;*

3. *$m(\mathcal{A}) = c$ if and only if $\mathbf{e}_k \in \mathcal{A}$ for $k \in [c]$, i.e., $\mathcal{A}$ contains the corner points of $\Delta_c$;*

4. *$m(\mathcal{A}) \le m(\mathcal{B})$ if $\mathcal{A} \subseteq \mathcal{B}$.*

We introduce next a predictive multiplicity metric called Rashomon Capacity that satisfies all properties above. In the rest of the paper, when the $\epsilon$-multiplicity set $\mathcal{M}_\epsilon(\cdot)$ is clear from context, we use $m(\mathbf{x}_i)$ as shorthand for $m(\mathcal{M}_\epsilon(\mathbf{x}_i))$.

## 3.2 Rashomon Capacity

Our goal is to quantify predictive multiplicity in terms of the score variations assigned to each sample $\mathbf{x}_i$ in a dataset $\mathcal{D}$, given a Rashomon set $\mathcal{R}(\mathcal{H}, \epsilon)$ and the corresponding $\epsilon$-multiplicity set $\mathcal{M}_\epsilon(\mathbf{x}_i)$. Note that an element in $\mathcal{M}_\epsilon(\mathbf{x}_i)$ is a probability distribution over $c$ classes. Thus, it is natural to adopt divergence measures for distributions to capture the "variation" of scores in $\mathcal{M}_\epsilon(\mathbf{x}_i)$. From a geometric viewpoint, a larger spread in scores indicates a greater amount of predictive multiplicity for a given sample $\mathbf{x}_i$.

Assume a probability measure (or "weight") $P_M$ across models in $\mathcal{R}(\mathcal{H}, \epsilon)$ (and therefore each score in $\mathcal{M}_\epsilon(\mathbf{x}_i)$), where $M$ denotes the random variable of selecting/sampling the models in the Rashomon set. Intuitively, if $P_M$ assigns mass 1 to a single model and 0 to all other models in the Rashomon set, then the output of only one model is considered. Alternatively, if $P_M$ is the uniform distribution, then the outputs of every model in the set are equally weighed. Given a divergence measure between distributions $d(\cdot\|\cdot)$, we quantify the spread of the scores in $\mathcal{M}_\epsilon(\mathbf{x}_i)$ by

$$\rho(\mathcal{M}_\epsilon(\mathbf{x}_i), P_M) \triangleq \inf_{\mathbf{q} \in \Delta_c} \mathbb{E}_{h \sim P_M} d(h(\mathbf{x}_i)\|\mathbf{q}). \tag{4}$$

Here, the minimizing $\mathbf{q}$ acts as a "center of gravity" or "centroid" for the outputs of the classifiers in the Rashomon set for a chosen distribution $P_M$ across models. Analogously, the quantity $\rho(\mathcal{M}_\epsilon(\mathbf{x}_i), P_M)$ can be understood as a measure of "spread" or "inertia" across model outputs. We select the distribution $P_M$ that results in the largest spread in scores:

$$C_d(\mathcal{M}_\epsilon(\mathbf{x}_i)) \triangleq \sup_{P_M} \rho(\mathcal{M}_\epsilon(\mathbf{x}_i), P_M) = \sup_{P_M} \inf_{\mathbf{q} \in \Delta_c} \mathbb{E}_{h \sim P_M} d(h(\mathbf{x}_i)\|\mathbf{q}). \tag{5}$$

The missing element is the choice of divergence measure $d(\cdot\|\cdot)$. A natural candidate is cross-entropy (or log-loss) $d(h(\mathbf{x}_i)\|\mathbf{q}) = -h(\mathbf{x}_i)^\top \log \mathbf{q}$, since this is the standard loss used for training and evaluating probabilistic classifiers. Alas, the minimal cross-entropy $\min_{\mathbf{q}} -h(\mathbf{x}_i)^\top \log \mathbf{q} = h(\mathbf{x}_i)^\top \log h(\mathbf{x}_i)$ is not 0 and depends on $h(\mathbf{x}_i)$. Consequently, if one were to choose $d(\cdot\|\cdot)$ to be cross-entropy, the minimum value of (5) would not be consistent and would depend on $\mathbf{x}_i$ — even when the outputs across all models in the Rashomon set match! Thus, we shift cross-entropy by its minimum $-h(\mathbf{x}_i)^\top \log h(\mathbf{x}_i)$. This results in KL-divergence as our divergence measure of choice: $D_{KL}(h(\mathbf{x}_i)\|\mathbf{q}) = -h(\mathbf{x}_i)^\top \log \mathbf{q} + h(\mathbf{x}_i)^\top \log h(\mathbf{x}_i)$. Putting it all together, we next formally define the spread in scores measured using KL-divergence as *Rashomon Capacity*.

---

[2]We exclude the empty set $\emptyset$ from $\mathcal{P}_c$, since the multiplicity of an empty scores set is not well-defined.
[3]We use the term "metric" loosely, not in the sense of defining a metric space over a set.

**Definition 2.** *Given a sample $\mathbf{x}_i$, a Rashomon set $\mathcal{R}(\mathcal{H}, \epsilon)$, and the corresponding $\epsilon$-multiplicity set $\mathcal{M}_\epsilon(\mathbf{x}_i)$, the Rashomon Capacity[4] is defined as*

$$m_C(\mathbf{x}_i) \triangleq 2^{C(\mathcal{M}_\epsilon(\mathbf{x}_i))}, \; \text{where } C(\mathcal{M}_\epsilon(\mathbf{x}_i)) = \sup_{P_M} \inf_{\mathbf{q} \in \Delta_c} \mathbb{E}_{h \sim P_M} D_{KL}(h(\mathbf{x}_i) \| \mathbf{q}), \quad (6)$$

*where the supremum in the right-hand side is taken over all probability measures $P_M$ over $\mathcal{R}(\mathcal{H}, \epsilon)$.*

The exponent $C(\mathcal{M}_\epsilon(\mathbf{x}_i))$ is ubiquitous in information theory; in fact, $C(\mathcal{M}_\epsilon(\mathbf{x}_i))$ is the *channel capacity* [14] of a channel $P_{Y|M}$ whose rows are the entries of $\mathcal{M}_\epsilon(\mathbf{x}_i)$. This connection motivates the name "Rashomon Capacity" and is useful for proving that $m_C(\mathbf{x}_i)$ is indeed a predictive multiplicity metric, as stated in the next proposition.

**Proposition 1.** *The function $m_C(\cdot) = 2^{C(\mathcal{M}_\epsilon(\cdot))} : \mathcal{X} \to [1, c]$ satisfies all properties of a predictive multiplicity metric in Definition 1.*

In contrast, ambiguity and discrepancy in (2) do not satisfy the properties of a predictive multiplicity metric outlined in Definition 1. An interesting connection between ambiguity and Rashomon Capacity is that ambiguity measures the fraction of samples in a dataset with non-zero Rashomon Capacity. In addition, Rashomon Capacity is fundamentally different from the size of a Rashomon set, in the sense that a larger Rashomon set does not necessarily lead to a larger Rashomon Capacity. Using a binary classification problem as an example, consider two Rashomon sets with scores

$$\mathcal{R}_1(\mathcal{H}, \epsilon) = \{h_1, h_2, h_3\}, \; h_1(\mathbf{x}_i) = [0.45, 0.55], h_2(\mathbf{x}_i) = [0.50, 0.50], h_3(\mathbf{x}_i) = [0.60, 0.40],$$
$$\mathcal{R}_2(\mathcal{H}, \epsilon) = \{h_1, h_2\}, \; h_1(\mathbf{x}_i) = [0.85, 0.15], h_2(\mathbf{x}_i) = [0.10, 0.90].$$
$$(7)$$

$\mathcal{R}_2(\mathcal{H}, \epsilon)$ has a larger Rashomon Capacity than $\mathcal{R}_1(\mathcal{H}, \epsilon)$, albeit $|\mathcal{R}_2(\mathcal{H}, \epsilon)| = 2 < |\mathcal{R}_1(\mathcal{H}, \epsilon)| = 3$.

### 3.3 Rashomon Capacity in score and decision domains

Rashomon Capacity is defined in terms of the raw outputs in $\Delta_c$ of a probabilistic classifier; therefore it can also be evaluated with decisions, since a decision, after one-hot encoding, still lies in the probability simplex (at a vertex). This allows Rashomon Capacity to provide a more nuanced view of score variation, and can be used to identify the number of "conflicting" classes in the predictions produced by models in the Rashomon set (i.e., Rashomon Capacity is between 1 and $c$).

Taking a ternary classification as an example, for three score vectors $[0.49, 0.51, 0]$ and $[0.51, 0.49, 0]$, the Rashomon Capacity of the raw scores for these samples is close to 1. For the thresholded scores ($[0, 1, 0]$ and $[1, 0, 0]$) the Rashomon Capacity is now 2. Since Rashomon Capacity is between 1 and 3, this indicates that the confusion is between 2 classes instead of 3. In contrast, prior metrics that also operate on thresholded scores, such as ambiguity and discrepancy (2), only capture agreement among predictions. In this sense, score-level metrics could potentially provide a finer characterization of predictive multiplicity. This *does not* mean that multiplicity should only be reported in the score domain. On the contrary, our suggestion is that multiplicity should be measured at both the score and threshold levels and reported to stakeholders, as scores and decisions paint different pictures on how models conflict.

## 4 Computational challenges of multiplicity metrics

The computation of any multiplicity metric requires an approximate characterization of the Rashomon set—even for simple hypothesis spaces such as logistic regression. For instance, the computation of the Rashomon ratio involves estimating $\text{vol}(\mathcal{R}(\mathcal{H}, \epsilon))$, which is a level set estimation problem [15], and is computationally infeasible[5] when the hypothesis space $\mathcal{H}$ is large [16]. For a logistic regression, the exact form of the pattern Rashomon ratio is not tractable due to the non-linearity of the maximum likelihood ratio [17]. Similarly, the computation of ambiguity/discrepancy requires solving an optimization over the Rashomon set and can be computationally burdensome since 0-1 loss is not differentiable.

---

[4]We consider logarithms in base 2, and the unit of $C(\mathcal{M}_\epsilon(\mathbf{x}_i))$ is bit. We include further discussions and a geometric interpretation of Rashomon Capacity in Sections SM. 2.3 and SM. 2.4 respectively.

[5]An exact computation of $\text{vol}(\mathcal{R}(\mathcal{H}, \epsilon))$ in a special case, e.g., ridge regression where the Rashomon set forms an ellipsoid, is provided in [3, Section 5.1].

Given a dataset and a hypothesis space, there are two core challenges in computing any predictive multiplicity metric (see Definition 1). The first challenge is selecting an appropriate Rashomon parameter $\epsilon$, since the smallest achievable test loss $L(h_\theta)$ is unknown and only empirically approximated using a dataset with finite samples. The second challenge is approximating the Rashomon set without exhaustively searching the hypothesis space[6]. Next, we discuss strategies for addressing these two challenges.

**Selection of the Rashomon parameter $\epsilon$.** The value of $\epsilon$ can be set relative to the performance of a reference model. A natural choice of the reference model is the empirical risk minimizer, i.e., $h_{\theta^*}$, where $\theta^* \in \arg\min_{\theta \in \Theta} \hat{L}(h_\theta)$ (see Section 2 for notation). Here, we can set $\epsilon = \hat{L}(h_{\theta^*}) + \epsilon'$ with $\epsilon' \geq 0$, i.e., the Rashomon parameter depends on the minimum empirical loss. The parameter $\epsilon'$, in turn, can be selected in terms of the upper boundary of a confidence interval around the empirical minimum. For example, the confidence interval can be estimated via *bootstrapping* (see Fig. SM. 4.6 for bootstrapped loss intervals). Naturally, $\epsilon'$ depends on the size of the test set used for evaluating $h_{\hat{\theta}}$—when the dataset has $n$ samples, usually $\epsilon'$ will be of the order $O(1/\sqrt{n})$ [19]. If more samples are available to evaluate model performance (rendering narrower confidence intervals), this parameter's value will decrease.

The discussion above motivates the one-sided definition for the Rashomon set in (1). If one were able to find the unique classifier that outperforms all others in terms of population loss, then it is justified to use this classifier in practice. In fact, this point is eloquently made in [5], which argues that there is no (moral) harm in using the most accurate classifier, i.e., selecting a classifier that has a provable smaller average loss (or higher accuracy) than another. Existing works on multiplicity, e.g., [3] and [4], also adopt a one-sided Rashomon set definition.

**The Rashomon subset.** With limited computational power and memory, exploring the Rashomon set, i.e., searching the hypothesis space $\mathcal{H}$ to find all models with test losses smaller than $\epsilon$, is challenging. For several hypothesis spaces (e.g., neural networks), we are only able to acquire a small number of models in the Rashomon set. We denote the collection of these models as a *Rashomon subset* $\widetilde{\mathcal{R}}(\mathcal{H}, \epsilon)$.

The Rashomon subset can be used to approximate the "true" Rashomon set $\mathcal{R}(\mathcal{H}, \epsilon)$ in the evaluation of multiplicity metrics. We can construct a Rashomon subset with $K$ models by choosing the Rashomon parameter $\epsilon$ in terms of a reference model $h_{\theta^*}$, i.e.,

$$\widetilde{\mathcal{R}}(\mathcal{H}, \epsilon') \triangleq \{h_{\theta_i} \in \mathcal{H}; L(h_{\theta_i}) \leq \hat{L}(h_{\theta^*}) + \epsilon'\}_{i=1}^K \subseteq \mathcal{R}(\mathcal{H}, \hat{L}(h_{\theta^*}) + \epsilon'). \tag{8}$$

For example, the Rashomon ratio can be approximated as $K/\text{vol}(\mathcal{H})$. Similarly, we can optimize for ambiguity/discrepancy in (2) over $h_{\theta_i} \in \widetilde{\mathcal{R}}(\mathcal{H}, \epsilon')$. Finally, we can approximate the $\epsilon'$-multiplicity subset $\widetilde{\mathcal{M}}_{\epsilon'}(\mathbf{x}_i) \triangleq \left\{ h_{\theta_i}(\mathbf{x}_i); h \in \widetilde{\mathcal{R}}(\mathcal{H}, \epsilon') \right\}$ in (3) for Rashomon Capacity.

In this sense, evaluating multiplicity metrics boils down to finding a Rashomon subset $\widetilde{\mathcal{R}}(\mathcal{H}, \epsilon')$ and computing the metric for that set. This is an approximation of the true multiplicity, yet becomes more accurate with an increasing $K$. Next, we provide an algorithm based on model weight perturbation to find a Rashomon subset for estimating Rashomon Capacity for any differentiable model.

Note that, unless the Rashomon set can be fully characterized, all estimates of multiplicity metrics are underestimates based on approximating the true Rashomon set by a Rashomon subset. We emphasize that identifying and disclosing predictive multiplicity — even if an underestimate — is still critical for models deployed in applications of individual-level consequence (e.g., healthcare, education, lending), and is better than the current practice reporting no multiplicity at all.

**Computing Rashomon Capacity.** The definition of Rashomon Capacity in (6) does not assume a finite cardinality of the Rashomon set. Remarkably, even when the Rashomon set has infinite cardinality, the value of Rashomon Capacity of a sample can be recovered by considering only a small number of models in the Rashomon set. In fact, for each sample $\mathbf{x}_i$, there exists a $\epsilon$-multiplicity subset of at most $c$ models that fully captures the variation in scores. This statement is formalized by the next proposition, which can be proven by applying Carathéodory's theorem [20].

---

[6]In neural networks, for example, the size of the Rashomon set is determined by the number of local minima, which grows exponentially many with the number of parameters [18].

**Proposition 2.** *For each sample $\mathbf{x}_i \in \mathcal{D}$, there exists a $\epsilon$-multiplicity subset $\widetilde{\mathcal{M}}_\epsilon(\mathbf{x}_i) \subseteq \mathcal{M}_\epsilon(\mathbf{x}_i)$ with $|\widetilde{\mathcal{M}}_\epsilon(\mathbf{x}_i)| \leq c$ that fully captures the spread in scores for $\mathbf{x}_i$ across the Rashomon set, i.e., $C(\widetilde{\mathcal{M}}_\epsilon(\mathbf{x}_i)) = C(\mathcal{M}_\epsilon(\mathbf{x}_i))$. In particular, there are at most $c$ models in a Rashomon subset $\widetilde{\mathcal{R}}(\mathcal{H}, \epsilon)$ whose output scores yield the same Rashomon Capacity for $\mathbf{x}_i$ as the entire Rashomon set.*

Proposition 2 implies that, for each sample, there exists a Rashomon subset with $c$ models that captures all score variation. In other words, the value of Rashomon Capacity remains the same regardless if we measure multiplicity on this subset or on the entire Rashomon set. This allows us to circumvent the task of characterizing the entire Rashomon set (which has potentially infinite models), and focus on identifying $c$ models per sample that maximize score variations while still satisfying a target loss constraint. We describe next a method (described in detail in Algorithm SM. 2) based on weight perturbation that obtains $c$ models in the Rashomon subset for each sample.

Given a sample $\mathbf{x}_i$, we obtain models with output predictions $\mathbf{p}_k$ by approximately solving the following optimization problem which maximizes the output score for class $k$:

$$\mathbf{p}_k = h_{\hat{\theta}}(\mathbf{x}_i), \text{ where } \hat{\theta} = \underset{h_\theta \in \mathcal{R}(\mathcal{H}, \epsilon)}{\arg \max} [h_\theta(\mathbf{x}_i)]_k, \ \forall k = 1, 2, \cdots, c. \tag{9}$$

To solve (9), for each $k$, we set the objective to be $\min_{\theta \in \Theta} -[h_\theta(\mathbf{x}_i)]_k$, compute the gradients, and update the parameter $\theta$ until $L(h_\theta) > \epsilon$. Given a pre-trained model in the Rashomon set, (9) can be viewed as an adversarial weight perturbations (AWP) technique to explore the Rashomon set [21, 22] (see Section SM. 2.6 for exact weight perturbation on logistic regression).

With the discrete $\epsilon$-multiplicity subset obtained by solving (9), Rashomon Capacity can be computed by standard procedures such as the Blahut–Arimoto (BA) algorithm [23, 24]. The BA algorithm is a class of iterative algorithms for numerically computing *discrete* channel capacity (or more generally, the rate-distortion function), see Section SM. 2.5 for more details. Note that AWP may still underestimate the true Rashomon Capacity, yet it greatly improves the estimates compared to prior work and is less computationally intensive than sampling, as shown in the next section.

The procedure in (9) reveals a desirable property of a Rashomon subset: it should include models with significant score variations. Similarly, a desirable Rashomon subset for accurately evaluating ambiguity/discrepancy is one with models that have most score disagreement. This property also explains why collecting a Rashomon subset by straightforwardly sampling models [3] in the Rashomon set could be inefficient (cf. Algorithm SM. 1) since the randomly sampled models would not necessarily have a significant score disagreement/variation. In fact, the sampling strategy could require a significant amount of models; see for example in [3, Section 6], a Rashomon subset consisting of 250k decision trees.

## 5 Empirical study

We illustrate how to measure, report, and potentially resolve predictive multiplicity of probabilistic classifiers using Rashomon Capacity on UCI Adult [25], COMPAS [26], HSLS [27], and CIFAR-10 datasets [8]. UCI Adult and COMPAS are two binary classification datasets on income and recidivism prediction, respectively, and are widely used in fairness research [28]. The HSLS is an education dataset, collected from high school students in the USA, whose features include student and parent information (see Section SM. 3.1 for details). We created a binary label $Y$ from students' $9^{\text{th}}$-grade math test scores (i.e., top 50% vs. bottom 50%). We select the first three datasets to illustrate the effect of predictive multiplicity on individuals. Finally, we include the CIFAR-10 dataset to demonstrate how to report Rashomon Capacity in multi-class classification.

For the classifiers, we adopt feed-forward neural networks for the first three datasets, and a convolutional neural network VGG16 [9] for CIFAR-10. For more information on the datasets, neural network architectures, and training details, see Section SM. 3.2. All numbers reported are evaluated on the test set.

**Measuring and reporting predictive multiplicity via Rashomon Capacity.** We evaluate two methods, `sampling` with different weight initialization seeds [3] and `AWP` (9), and report the Rashomon Capacity in Fig. 3. For `sampling`, we construct a Rashomon subset $\widetilde{\mathcal{R}}(\mathcal{H}, \epsilon')$ with 100 models by different random initialization (cf. (8) with $K = 100$). For `AWP`, the $\epsilon'$-multiplicity subsets $\widetilde{\mathcal{M}}_{\epsilon'}(\mathbf{x}_i)$

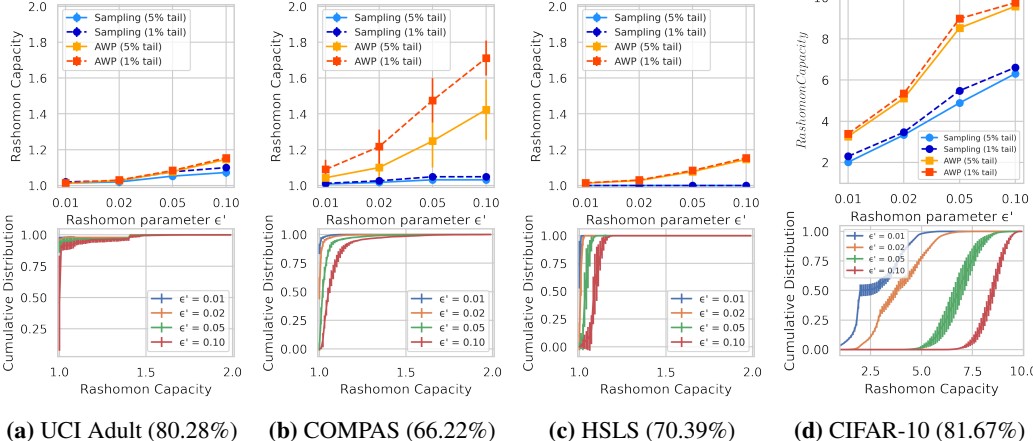

**(a)** UCI Adult (80.28%)  **(b)** COMPAS (66.22%)  **(c)** HSLS (70.39%)  **(d)** CIFAR-10 (81.67%)

**Figure 3:** For each dataset (percentage is test accuracy), the top figure shows the mean and standard error of the largest 1% and 5% (1% tail and 5% tail in the legend) Rashomon Capacity among all the samples with difference Rashomon parameter $\epsilon$. Two methods are used to obtain models from the Rashomon set, AWP (9) and random sampling. The bottom figure shows the cumulative distribution of the Rashomon Capacity of all the samples obtained by AWP. Each point is generated with 5 repeated splits of the dataset.

of each sample $\mathbf{x}_i$ is constructed by the $\mathbf{p}_1, \cdots, \mathbf{p}_c$ obtained from (9). For example, when $\epsilon' = 0.01$, the models in the Rashomon set $\widetilde{\mathcal{R}}(\mathcal{H}, 0.01)$ achieves small and statistically indistinguishable test losses from each other; however, the Rashomon Capacity is non-zero for a significant fraction of samples. In particular, we showcase the average top 1% and top 5% of the Rashomon Capacity from all samples for different $\epsilon'$, and the cumulative distribution of the Rashomon Capacity across the samples. As the Rashomon parameter increases, both sampling and AWP lead to higher Rashomon Capacity since the Rashomon set gets larger. Observe that the increase of Rashomon Capacity when $\epsilon'$ varies from 0.01 to 0.02 is different for UCI Adult and COMPAS datasets, since the choice of $\epsilon'$ is data-dependent. The AWP (9) achieves higher Rashomon Capacity than random sampling as AWP intentionally explores the Rashomon set that maximizes the scores variations. It is important to keep in perspective that *each sample* in the high-Rashomon Capacity tail displayed in Fig. 3 corresponds to an *individual* who receives conflicting predictions. In applications such as criminal justice and education, conflicting predictions for even one individual should be reported in, e.g., model cards [13].

For the experiments in Figure 3, the estimated Rashomon ratio are almost all zeros since the hypothesis space, parameterized by millions of parameters, is always significantly larger. In SM, we further report (i) other strategies to explore the Rashomon set in Section SM. 4.2, (ii) Rashomon Capacity evaluated with decisions instead of scores in Section SM. 4.3, and (iii) other metrics of multiplicity such as ambiguity/discrepancy in Section SM. 4.4.

**Resolving predictive multiplicity by greedy model selection.** Predictive multiplicity could be resolved by, for example, selecting a subset of models (potentially of size one in the ideal case) and releasing the scores to a stakeholder. We propose a greedy model selection procedure to select a subset of competing classifiers for resolving predictive multiplicity. Given $R$ competing classifiers, the goal is to select $r$ models ($r < R$) that result in distributions of the Rashomon Capacity similar to that of the original $R$ models. Starting from a dataset $\mathcal{D}$ and a Rashomon subset $\widetilde{\mathcal{R}}(\mathcal{H}, \epsilon')$, this can be implemented by (i) initializing a set $\mathcal{A}$ of models by randomly selecting a model in $\widetilde{\mathcal{R}}(\mathcal{H}, \epsilon')$, (ii) growing $\mathcal{A}$ by adding one model from $\widetilde{\mathcal{R}}(\mathcal{H}, \epsilon')$ that maximizes the average Rashomon Capacity across $\mathcal{D}$, and (iii) stopping until there are $r$ models in $\mathcal{A}$. This greedy model selection is inspired by Property 4 (monotonicity) in Definition 1, since including the models to the set $\mathcal{A}$ does not reduce capacity. In Fig. 4, we models from the Rashomon sets for UCI Adult, COMPAS, HSLS and CIFAR-10 datasets respectively. Here, the hypothesis space are feed-forward neural networks (see details in Section SM. 3.2). Observe that only a small subset of the sampled models, selected by the greedy model selection procedure, is required to recover the distribution of the Rashomon Capacity. On COMPAS dataset, the 10 models obtained by the greedy model selection procedure capture the Rashomon Capacity computed with the original 163 models, i.e., these 10 models display most of the score variations.

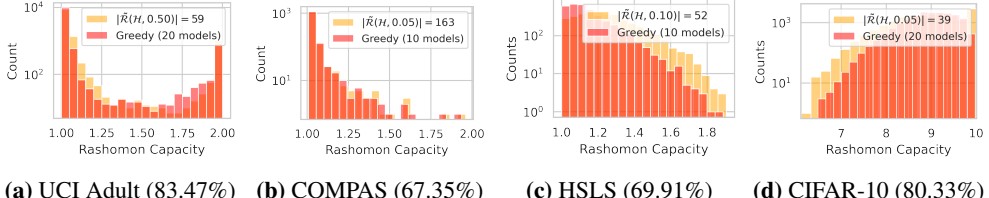

**(a)** UCI Adult (83.47%)  **(b)** COMPAS (67.35%)  **(c)** HSLS (69.91%)  **(d)** CIFAR-10 (80.33%)

**Figure 4:** The distributions of the Rashomon Capacity for UCI Adult, COMPAS, HSLS and CIFAR-10 datasets ($\epsilon$ is percentage of mean test accuracy) obtained by sampling models from the Rashomon subset and applying greedy model selection procedure (Greedy in the legend) on the sampled models.

In Section SM. 4.1, we provide a worked-out example with the COMPAS dataset on how Rashomon Capacity can be used to identify high-multiplicity samples. We observe patterns in sex and prior convictions leading to samples having high Rashomon Capacity. We further discuss promising strategies, e.g., ensemble methods, model calibration and weight regularization to resolve multiplicity, in Sections SM. 4.5 and SM. 4.6. We observe that the ensemble method could lead to a smaller Rashomon Capacity, and is a viable strategy for resolving multiplicity in small models, but may be infeasible for large, computationally expensive models. In addition, weight regularization for logistic regression (e.g., LASSO or ridge penalties) could also reduce Rashomon Capacity. On the other hand, a perfectly calibrated classifier does not necessarily resolve multiplicity—a classifier whose predicted classes matches the true classes "on average" across samples does not necessarily translate to a consistent set of predictions for a single target sample across equally calibrated classifiers.

## 6   Final remarks

**Limitations.**   The AWP (9), despite being more efficient than random sampling, still requires re-training/perturbing a significant amount of models, and is computationally burdensome when scaling up to large datasets with millions of samples. Rashomon Capacity in certain cases may seem small for already significant score variations across classes, due to the convexity of KL-divergence. Indeed, this issue will occur for any strictly convex measure of divergence in (4). We provide a further discussion of how to interpret the numerical values of Rashomon Capacity in Section SM. 2.7.

**Future directions.**   First, overcoming the computational bottleneck to efficiently explore the Rashomon set is an impactful direction for optimization techniques. Second, Rashomon Capacity could be generalized to other probability divergences, e.g., $f$-divergences [29], Rényi divergence [30], or Wasserstein distance [31]. This generalization could potentially provide further operational significance and tunability for measuring multiplicity, as long as the conditions in Definition 1 are satisfied. Third, ensemble methods could be a promising strategy to reduce predictive multiplicity that worth studying.

**Broader impacts.**   The Rashomon effect impacts model selection [32–35], explainability [36], and fairness [37]. Rudin et al. [32] suggested that, given the choice of competing models, machine learning practitioners should select interpretable models *a priori*, rather than selecting a black-box model with conjectural explanations post-training. Hancox-Li [33] and D'Amour et al. [34] argued that epistemic patterns, e.g., causality, should be reflected when selecting models the Rashomon set. Black et al. [35] further studied multiplicity in the context of the conventional bias-variance trade-off analysis. Competing models in the Rashomon set may not only render conflicting explanations for predictions [36] and measures of feature importance [2], but also have inconsistent performance across population sub-groups. Consequently, the arbitrary choice of a single model may result in unnecessary and discriminatory bias against vulnerable population groups [37]. See Section SM. 2.1 for further discussion, including connections with individual fairness [38].

## Acknowledgement

This material is based upon work supported by the National Science Foundation under grants CAREER 1845852, IIS 1926925, and FAI 2040880, and by Meta Ph.D. fellowship.

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
