# Supplementary Materials
# Rashomon Capacity: A Metric for
# Predictive Multiplicity in Classification

**Hsiang Hsu and Flavio P. Calmon**

John A. Paulson School of Engineering and Applied Sciences, Harvard University

hsianghsu@g.harard.edu, flavio@seas.harvard.edu

This supplementary materials include omitted proofs for Proposition 1 and 2, additional explanations and discussions, details on experiment setups and training, and additional experiments. For clarity, the numbers with a prefix *SM.* refer to equations, figures, and tables in the supplementary material; numbers without the prefix refer to equations, figures, and tables in the main paper.

## SM. 1    Omitted proofs

### SM. 1.1    Proof of Proposition 1

**Proposition 1.** *The function $m_C(\cdot) = 2^{C(\mathcal{M}_\epsilon(\cdot))} : \mathcal{X} \to [1, c]$ satisfies all properties of a predictive multiplicity metric in Definition 1.*

*Proof.* For clarity, we assume $|\mathcal{M}_\epsilon(\mathbf{x}_i)| = m$. By the information inequality [1, Theorem 2.6.3] the mutual information $I(M; Y)$ between the random variables $M$ and $Y$ (defined in Section 3) is non-negative, i.e., $I(M; Y) \geq 0$. Moreover, since $I(M; Y) = H(Y) - H(Y|M) \leq \log c$, we have $0 \leq C(\mathcal{M}_\epsilon(\mathbf{x}_i)) \leq \log c$, and therefore $1 \leq 2^{C(\mathcal{M}_\epsilon(\mathbf{x}_i))} \leq c$ (since we pick the log base to be 2). If all rows in the transition matrix are identical, $Y$ is independent of $M$, and $H(Y|M) = \sum_{j=1}^m p_m(j) H(Y|M = j) = \sum_{j=1}^m p_m(j) H(Y) = H(Y)$; thus, $C(\mathcal{M}_\epsilon(\mathbf{x}_i)) \leq H(Y) - H(Y|M) = 0$ and $m_C(\mathbf{x}_i) = 2^{C(\mathcal{M}_\epsilon(\mathbf{x}_i))} = 1$. On the other hand, we denote the $c$ models in $\mathcal{R}(\mathcal{H}, \epsilon)$ which output scores are the "vertices" of $\Delta_c$ to be $m_1, \cdots, m_c$, then $H(Y|M = m_k) = 0, \ \forall k \in [c]$. $H(Y|M)$ is minimized to 0 by setting the weights $p_m$ on those c models to be $\frac{1}{c}$ and the rest to be 0. Thus, $C(\mathcal{M}_\epsilon(\mathbf{x}_i)) = H(Y) - H(Y|M) = H(Y)$ and $m_C(\mathbf{x}_i) = 2^{C(\mathcal{M}_\epsilon(\mathbf{x}_i))} = c$. Finally, let $\mathcal{M}_\epsilon^1(\mathbf{x}_i) \subseteq \mathcal{M}_\epsilon^2(\mathbf{x}_i)$ with random variables $M_1$ and $M_2$ respectively. Without loss of generality, assume that $\mathcal{M}_\epsilon^1(\mathbf{x}_i) = \{h_1(\mathbf{x}_i), \cdots, h_r(\mathbf{x}_i)\}$ and $\mathcal{M}_\epsilon^2(\mathbf{x}_i) = \mathcal{M}_\epsilon^1(\mathbf{x}_i) \cup \{h_{r+1}(\mathbf{x}_i)\}$, and we have

$$
\begin{aligned}
\mathbb{E}_{h \sim P_{M_2}} D(h(\mathbf{x}_i) \| \mathbf{q}) &= \sum_{i=1}^{r+1} P_{M_2}(h_i) D_{KL}(h_i(\mathbf{x}_i) \| \mathbf{q}) \\
&= \sum_{i=1}^{r} P_{M_2}(h_i) D_{KL}(h_i(\mathbf{x}_i) \| \mathbf{q}) + P_{M_2}(h_{r+1}) D_{KL}(h_{r+1}(\mathbf{x}_i) \| \mathbf{q}) \quad \text{(SM. 1)} \\
&\geq \sum_{i=1}^{r} P_{M_2}(h_i) D_{KL}(h_i(\mathbf{x}_i) \| \mathbf{q}) = \mathbb{E}_{h \sim P_{M_1}} D(h(\mathbf{x}_i) \| \mathbf{q}).
\end{aligned}
$$

36th Conference on Neural Information Processing Systems (NeurIPS 2022).

Therefore, $I(M_1; Y) = \inf_{P_{M_1}} \mathbb{E}_{h \sim P_{M_1}} D(h(\mathbf{x}_i) \| \mathbf{q}) \leq \inf_{P_{M_2}} \mathbb{E}_{h \sim P_{M_2}} D(h(\mathbf{x}_i) \| \mathbf{q}) = I(M_2; Y)$, and

$$I(M_1; Y) \leq I(M_2; Y) \Rightarrow \sup_{M_1} I(M_1; Y) \leq \sup_{M_2} I(M_2; Y)$$
$$\Rightarrow C(\mathcal{M}_\epsilon^1(\mathbf{x}_i)) \leq C(\mathcal{M}_\epsilon^2(\mathbf{x}_i)) \quad \text{(SM. 2)}$$
$$\Rightarrow m_C(\mathcal{M}_\epsilon^1(\mathbf{x}_i)) \leq m_C(\mathcal{M}_\epsilon^2(\mathbf{x}_i)),$$

since the power function is monotonic.

We now prove the converse statements. Assume $m(\mathbf{x}_i) = c$ and, thus, $C(\mathcal{M}_\epsilon(\mathbf{x}_i)) = \log c$. Let $P_M$ be the capacity-achieving distribution over models. Then $I(M; Y) = \log c$ and, from non-negativity of entropy and the fact that the uniform distribution maximizes entropy, $H(Y) = c$ and $H(Y|M) = 0$. Consequently, again from non-negativity of entropy, $H(Y|M = m) = 0$ for all $m \in \text{supp}(P_M)$, and thus $P_{Y|M=m}$ is an indicator function (i.e., given $M$, $Y$ is constant w.p.1). Since $H(Y) = c$, the result follows.

Finally, let $C(\mathcal{M}_\epsilon(\mathbf{x}_i)) = 0$ and $P_M$ be the capacity-achieving input distribution. Then $Y$ and $M$ are independent and, thus, $P_{Y|M=m} = P_Y$ (i.e., all scores are identical) for all values $m \in \text{supp}(P_M)$,. Since this holds for the capacity-achieving $P_M$, which in turn is the maximimum across input distributions, the converse result follows. □

## SM. 1.2   Proof of Proposition 2

**Proposition 2.** *For each sample $\mathbf{x}_i \in \mathcal{D}$, there exists a subset $\mathcal{A} \subseteq \mathcal{M}_\epsilon(\mathbf{x}_i)$ with $|\mathcal{A}| \leq c$ that fully captures the spread in scores for $\mathbf{x}_i$ across the Rashomon set, i.e., $m_C(\mathbf{x}_i) = 2^{C(\mathcal{A})}$. In particular, there are at most $c$ models in $\mathcal{R}(\mathcal{H}, \epsilon)$ whose output scores yield the same Rashomon Capacity for $\mathbf{x}_i$ as the entire Rashomon set.*

*Proof.* Carathéodory's theorem [2] states that if a point $x$ of $\mathbb{R}^d$ lies in the convex hull of a set $\mathcal{X}$, then x can be written as the convex combination of at most $d + 1$ points in $\mathcal{X}$. Namely, there is a subset $\mathcal{X}'$ of $\mathcal{X}$ consisting of $d + 1$ or fewer points such that $x$ lies in the convex hull of $\mathcal{X}'$.

In our case, we consider the random variable $M$ of the Rashomon set $\mathcal{R}(\mathcal{H}, \epsilon)$ and $Y = \Delta_c \triangleq \{\mathbf{g} \in \mathbb{R}^c; \sum_{i=k}^c [\mathbf{g}]_k = 1, \forall k \ [\mathbf{g}]_k \geq 0\}$ is a $(c-1)$ dimensional space[1]. We assume $|\mathcal{R}(\mathcal{H}, \epsilon)| = m$ in the following proof, but $\mathcal{R}(\mathcal{H}, \epsilon)$ could contain arbitrary large (or infinite) amount of output scores from the models in the Rashomon set. There are also m output scores $\{h_1(\mathbf{x}_i), \cdots, h_m(\mathbf{x}_i)\} \in \Delta_c$ in the $\epsilon$-multiplicity set $\mathcal{M}_\epsilon(\mathbf{x}_i)$ for each sample $\mathbf{x}_i \in \mathcal{D}$. By Carathéodory's theorem, since $\Delta_c$ is $(c-1)$ dimensional and is convex, any score $h(\mathbf{x}_i)$ can be expressed by the convex combination of $(c-1)+1 = c$ scores. Moreover, let the c scores be $\{h_1(\mathbf{x}_i), \cdots, h_c(\mathbf{x}_i)\}$, since Rashomon Capacity measures the spread of the scores, adding any score $h(\mathbf{x}_i) \in \text{convexhull}(h_1(\mathbf{x}_i), \cdots, h_c(\mathbf{x}_i))$ to the channel constructed by $\{h_1(\mathbf{x}_i), \cdots, h_c(\mathbf{x}_i)\}$ would not affect Rashomon Capacity. □

## SM. 2   Additional details

### SM. 2.1   Predictive multiplicity: fairness, reproducibility, and security

Predictive multiplicity and the Rashomon effect are related to individual fairness [3,4]. A mechanism $M : \mathcal{X} \to \mathcal{Y}$ satisfies individual fairness if for every $x, x' \in \mathcal{X}$, $D(M(x), M(x')) \leq d(x, x')$, where $d$ and $D$ are metrics on $\mathcal{X}$ and $\mathcal{Y}$ respectively. It is also called the $(D, d)$-Lipschitz property. Individual fairness aims to ensure that "similar individuals are treated similarly." The consequence of predictive multiplicity is that *the same* individual can be treated differently due to arbitrary and unjustified choices made during the training process (e.g., parameter initialization, random seed, dropout probability, etc.). Integrating predictive multiplicity and individual fairness could results in a more thorough formulation for fair machine learning.

Predictive multiplicity allows different predictions from competing classifiers for the samples. Thus predictive multiplicity could lead to different decision regions when training with the same dataset

---

[1]In the main paper, we say $\Delta_c$ is a c-dimensional probability simplex, but it does not mean $\Delta_c$ is a c dimensional space. In fact, $\Delta_c$ is a $(c-1)$ dimensional space.

and achieving similar performance, and makes it hard for a machine learning practitioner to reproduce the decision regions if a different initalization of a classifier is selected. Somepalli et al. [5] studied the reproducibility of decision regions of almost-equally performing learning models, and observe that changes in model architecture (which reflect the inductive bias) lead to visible changes in decision regions. Notably, neural networks with very narrows or wide layers have better reproducibility in their decision regions. On the other hand, neural networks with "moderate" number of neurons in each layer have decision regions fragmented into many small pieces, and are harder to reproduce. The connection between predictive multiplicity, neural network architectures, and inductive bias is also an interesting research direction. For example, a stronger inductive bias could restrict the arbitrariness of a training process, leading to smaller predictive multiplicity.

The fact that multiple classifiers may yield distinct predictions to a target a sample while having statistically identical average loss performance can also cause security issues in machine learning. The score variation could result from a malicious learner/designer who either plants an undetectable backdoor or carefully selects a specific model. This may result in intentional manipulation of the output scores for a sample without detectable performance changes [6].

## SM. 2.2   Predictive multiplicity with small Rashomon parameters

Note that $\epsilon = 0$ implies prefect generalization to the test set, and is in general infeasible due to a limited number of samples and optimization techniques. Moreover, a small $\epsilon$ could lead to high predictive multiplicity. Consider a classification task with 1000 samples with binary classes, and trained with the 0-1 loss. Suppose $\epsilon = 0.001$, i.e., only allowing one sample $\mathbf{x}_i$ at a time to be misclassified. If the hypothesis space is tremendous, it is possible to find a classifier that only assign the wrong label to any $\mathbf{x}_i$, and thus the ambiguity in (2) could be 1.

## SM. 2.3   Metrics for the spread of scores

The divergence measure between two distributions used in (4) is not restricted to KL-divergence. For example, given a convex function $f : (0, \infty) \to \mathbb{R}$ satisfying $f(1) = 0$, and assume that $P$ and $Q$ are two probability distributions over a set $\mathcal{X}$, and $P$ is absolutely continuous with respect to $Q$. The $f$-divergence between $P$ and $Q$ is given by [7]

$$D_f(P\|Q) \triangleq \mathbb{E}_Q\left[f\left(\frac{P(X)}{Q(X)}\right)\right]. \tag{SM. 3}$$

Different choices of $f$ lead to different divergence; for example, if $f(t) = t \log t$, $D_f(P\|Q) = D_{KL}(P\|Q)$; if $f(t) = (t - 1)^2$, $D_f(P\|Q) = \chi^2(P\|Q)$ is the chi-square divergence; if $f(t) = t \log t - (1 + t) \log(1 + t)/2$, $D_f(P\|Q) = D_{JS}(P\|Q)$ is the Jensen-Shannon divergence. Another example of a tunable probability divergence is the Rényi divergence $R_\alpha(P\|Q)$ of order $\alpha \in \mathbb{R}^+/\{1\}$, defined as [8]

$$D_\alpha(P\|Q) \triangleq \frac{1}{\alpha - 1} \log\left(\sum_x \left(\frac{P(x)}{Q(x)}\right)^\alpha Q(x)\right), \tag{SM. 4}$$

Its continuous extensions for $\alpha = 1$ and $\infty$ can also be defined. In particular, for $\alpha = 1$, the Rényi divergence recovers KL divergence, and for $\alpha = \infty$, $D_\infty(P\|Q) = \max_x \log P(x)/Q(x)$ is called the max-divergence. Both the $f$-divergence and Rényi divergence generalize the usual notion of KL-divergence used in this paper, and these families of divergences could also be used to measure the spread of the scores in the probability simplex. For example, Nielsen et al. [9] reported an iterative algorithm to numerically compute a centroid for a set of probability densities measured by the Jensen–Shannon divergence. However, these generalizations of the KL divergence do not necessarily lead to multiplicity metrics that satisfy the properties outlined in Definition 1. More importantly, when taking the supremum over the input distributions (see (5)), we are unaware of a procedure as simple as the Blahut-Arimoto algorithm to estimate the corresponding Rashomon Capacity if the probability divergence is not the KL-divergence. Exploring alternative metrics for measuring score "spread" is a promising future research direction.

## SM. 2.4   Geometric interpretation of Rashomon Capacity

In Section 3, we introduce the Rashomon capacity to measure the spread of scores from a geometric viewpoint. Here, we further discuss the pleasing geometric interpretations possessed by Rashomon

Capacity, which can be found in information theory. Particularly, given a sample $\mathbf{x}_i$, let the information radius $\mathsf{rad}(\mathcal{M}_\epsilon(\mathbf{x}_i))$ and information diameter $\mathsf{diam}(\mathcal{M}_\epsilon(\mathbf{x}_i))$ of the $\epsilon$-multiplicity set $\mathcal{M}_\epsilon(\mathbf{x}_i)$ be [10]

$$\mathsf{rad}(\mathcal{M}_\epsilon(\mathbf{x}_i)) = \inf_{\mathbf{q}\in\Delta_c} \sup_{\mathbf{p}\in\mathcal{M}_\epsilon(\mathbf{x}_i)} D_{KL}(\mathbf{p}\|\mathbf{q}), \; \mathsf{diam}(\mathcal{M}_\epsilon(\mathbf{x}_i)) = \sup_{\mathbf{p},\mathbf{p}'\in\mathcal{M}_\epsilon(\mathbf{x}_i)} D_{KL}(\mathbf{p}\|\mathbf{p}'),$$
(SM. 5)

we have

$$C(\mathcal{M}_\epsilon(\mathbf{x}_i)) \leq \mathsf{rad}(\mathcal{M}_\epsilon(\mathbf{x}_i)) \leq \mathsf{diam}(\mathcal{M}_\epsilon(\mathbf{x}_i)), \tag{SM. 6}$$

where $C(\mathcal{M}_\epsilon(\mathbf{x}_i)) = \mathsf{rad}(\mathcal{M}_\epsilon(\mathbf{x}_i))$ if $\mathcal{M}_\epsilon(\mathbf{x}_i)$ is a convex set [11].

The proof of (SM. 6) is straightforward:

$$
\begin{aligned}
C(\mathcal{M}_\epsilon(\mathbf{x}_i)) &= \sup_{\boldsymbol{\alpha}\in\Delta_m} \inf_{\mathbf{q}\in\Delta_c} \sum_{j=1}^m \alpha_j D_{KL}(\mathbf{p}_j\|\mathbf{q}) \\
&\leq \inf_{\mathbf{q}\in\Delta_c} \sup_{\boldsymbol{\alpha}\in\Delta_m} \sum_{j=1}^m \alpha_j D_{KL}(\mathbf{p}_j\|\mathbf{q}) \\
&= \inf_{\mathbf{q}\in\Delta_c} \sup_{\boldsymbol{\alpha}\in\Delta_m} \mathbb{E}_{m\sim\boldsymbol{\alpha}} D_{KL}(P_{Y|M=m}\|\mathbf{q}) \\
&\leq \inf_{\mathbf{q}\in\Delta_c} \sup_{\mathbf{p}\in\mathcal{M}_\epsilon(\mathbf{x}_i)} D_{KL}(\mathbf{p}\|\mathbf{q}) \triangleq \mathsf{rad}(\mathcal{M}_\epsilon(\mathbf{x}_i)) \\
&\leq \sup_{\mathbf{p},\mathbf{p}'\in\mathcal{M}_\epsilon(\mathbf{x}_i)} D_{KL}(\mathbf{p}\|\mathbf{p}') \triangleq \mathsf{diam}(\mathcal{M}_\epsilon(\mathbf{x}_i)).
\end{aligned}
$$
(SM. 7)

At first glance, the information radius or diameter seem to be more intuitive metrics to measure the "spread" of the all possible scores from the Rashomon set; however, both of them do not satisfy the properties in Definition 1. More importantly, (SM. 6) shows that Rashomon capacity is a tighter metric, and is less likely to overestimate the spread of scores, i.e., the predictive multiplicity. Moreover, maximizing the KL divergence is in general an ill-posed problem since the KL divergence is (jointly) convex, and could diverge to infinity. Zhang et al. [12] demonstrated that maximizing the KL divergence between the scores generated by a classifier with two different samples is solvable if the Euclidean distance between the two samples is upper bounded. In our case, we do not have control over $\mathbf{p}, \mathbf{p}' \in \mathcal{M}_\epsilon(\mathbf{x}_i)$ and the underlying models that output $\mathbf{p}, \mathbf{p}'$ since two models could be very different from each other (in terms of, e.g., the Euclidean distance of the model parameters), but still yield similar test loss due to the existence of multiple local minima.

## SM. 2.5 The Blahut-Arimoto algorithm

For the sake of completeness, we describe the Blahut-Arimoto (BA) algorithm [13, 14] used in Section 5 for computing channel capacity. For a discrete memoryless channel (DMC) $X \to Y$ with transition probabilities $P_{Y|X}$ and input probability $Q$, where $\mathcal{X} = [1, \cdots, m]$ and $\mathcal{Y} = [1, \cdots, c]$. The mutual information $I(X; Y)$ between $X$ and $Y$ is defined as

$$I(X;Y) \triangleq \sum_{i=1}^m \sum_{j=1}^c P_{X,Y}(i,j) \log \frac{P_{X,Y}(i,j)}{Q(i)P_Y(j)} = \sum_{i=1}^m \sum_{j=1}^c P_{Y|X}(j|i)Q(i) \log \frac{P_{X|Y}(i|j)}{Q(i)}. \tag{SM. 8}$$

By definition, the capacity of the channel $P_{Y|X}$ is defined as

$$C(P_{Y|X}) = \max_Q I(X;Y) = \max_Q \sum_{i=1}^m \sum_{j=1}^c P_{Y|X}(j|i)Q(i) \log \frac{P_{X|Y}(i|j)}{Q(i)}, \tag{SM. 9}$$

where $P_{X|Y}(i|j) = \frac{P_{Y|X}(j|i)Q(i)}{\sum_k P_{Y|X}(j|k)Q(k)}$. Since $P_{X|Y}(i|j)$ can be viewed as a function of the channel $P_{Y|X}$ and $Q$, from (SM. 9), it is clear that for a fixed channel $P_{Y|X}$, the channel capacity is a convex function of the input probabilities $Q$. Denote any $P_{X|Y}(i|j) = \Phi(i|j)$, we can alternatively express the mutual information as

$$I(X;Y) = \sum_{i=1}^m \sum_{j=1}^c P_{Y|X}(j|i)Q(i) \log \frac{\Phi(i|j)}{Q(i)} = J(Q, \Phi). \tag{SM. 10}$$

It can be proven that [13, 14]

1. For a fixed $Q$, $J(Q, \Phi) \leq J(Q, P_{X|Y})$, i.e., $J(Q, P_{X|Y}) = \max_\Phi J(Q, \Phi)$, and therefore $C(P_{Y|X}) = \max_Q \max_\Phi J(Q, \Phi)$.

2. For a fixed $\Phi$, $J(Q, \Phi) \leq \log\left(\sum_{i=1}^m r(i)\right)$, $r(i) = \exp\left[\sum_{j=1}^c P_{Y|X}(j|i) \log \Phi(i|j)\right]$, where equality holds if and only if $Q(i) = r(i)/\sum_{k=1}^m r(k)$.

The BA algorithm is built upon these two properties, and doubly maximizes $J(Q, \Phi)$. More specifically, let $t$ be the iteration index, and let $Q^0$ be a choose initialization of the input distribution, for each iteration, we update $\Phi$ and $Q$ by

1. $\Phi^{l+1}(i|j) = \frac{Q^l(i)P_{Y|X}(j|i)}{\sum_{k=1}^m Q^l(k)P_{Y|X}(j|k)}$, $\forall i, j$.

2. $r^{l+1}(i) = \exp\left(\sum_{j=1}^c P_{Y|X}(j|i) \log \Phi^{l+1}(i|j)\right)$.

3. $Q^{l+1}(i) = \frac{r^{l+1}(i)}{\sum_{k=1}^m r^{l+1}(k)}$.

4. $J(Q^{l+1}, \Phi^{l+1}) = \log\left(\sum_{i=1}^m r^{l+1}(i)\right)$.

5. $l = l + 1$.

For the stopping criteria, let $c^l(i) = r^l(i)/Q^l(i)$, we have $J(Q^l, \Phi^l) = \log\left(\sum_{i=1}^m Q^l(i)c^l(i)\right)$. Since $J(Q^l, \Phi^l)$ is the logarithm of the average of $c^l(i)$, we have

$$\log\left(\sum_{i=1}^m Q^l(i)c^l(i)\right) \leq C(P_{Y|X}) \leq \max_i \log c^l(i), \tag{SM. 11}$$

and therefore we update $Q^{l+1}(i)$ and $\Phi^{l+1}(i|j)$ until the stopping criteria is matched,

$$\max_i \log c^l(i) - \log\left(\sum_{i=1}^m Q^l(i)c^l(i)\right) \leq \epsilon, \tag{SM. 12}$$

where $\epsilon > 0$ is a pre-defined accuracy parameter.

The BA algorithm has also been extended to channels with continuous input and output alphabets, i.e., $|\mathcal{X}| = \infty$ and $|\mathcal{Y}| = \infty$, based on sequential Monte-Carlo integration methods (i.e., particle filters) [15–18]. Since we deal with finite predicted classes and discrete Rashomon sets, Proposition 2 allows us to circumvent the use of more sophisticated variations of the BA algorithm.

### SM. 2.6 Adversarial weight perturbation on unregularized logistic regression

In (9), we introduce an adversarial weight perturbation procedure to estimate Rashomon Capacity in the Rashomon set. In general, the problem in (9) is difficult to analyze, and is usually optimized by using automated gradient computation tools such as Tensorflow [19]. Here, we provide a special case of unregularized logistic regression, which gradient and Hessian can be analytically computed. We start with a more general case by considering a set of features and labels $\{\mathbf{z}_i, y_i\}_{i=1}^n$ for a binary classification problem, where $\mathbf{z}_i \in \mathbb{R}^m$ and $y_i \in \{0, 1\}$. Logistic regression assumes the output scores

$$P(\hat{Y} = 1 | Z = \mathbf{z}_i; \mathbf{w}) = \frac{e^{\mathbf{z}_i^\top \mathbf{w}}}{1 + e^{\mathbf{z}_i^\top \mathbf{w}}} \text{ and}$$

$$P(\hat{Y} = 0 | Z = \mathbf{z}_i; \mathbf{w}) = 1 - P(\hat{Y} = 1 | Z = \mathbf{z}_i; \mathbf{w}) = \frac{1}{1 + e^{\mathbf{z}_i^\top \mathbf{w}}}, \tag{SM. 13}$$

where $\mathbf{w} \in \mathbb{R}^m$ is the vector of weights. The loss in logistic regression (without regularization) is defined as [20]

$$\ell(\mathbf{w}) = -\sum_{i=1}^n \left(y_i \log P(\hat{Y} = 1 | Z = \mathbf{z}_i; \mathbf{w}) + (1 - y_i) \log(1 - P(\hat{Y} = 1 | Z = \mathbf{z}_i; \mathbf{w}))\right)$$

$$= \sum_{i=1}^n \left(y_i \mathbf{w}^\top \mathbf{z}_i - \log(1 + e^{\mathbf{w}^\top \mathbf{z}_i})\right). \tag{SM. 14}$$

Let $\mathbf{Z} = [\mathbf{z}_1, \cdots, \mathbf{z}_n]^\top \in \mathbb{R}^{n \times m}$ be the feature matrix, $\mathbf{y} = [y_1, \cdots, y_n]^\top$ the label vector, $\mathbf{p} = [P(\hat{Y} = 1 | Z = \mathbf{z}_1; \mathbf{w}), \cdots, P(\hat{Y} = 1 | Z = \mathbf{z}_n; \mathbf{w})]$ be the score vector, and $\mathbf{W} = \mathrm{diag}(\mathbf{w}_1, \cdots, \mathbf{w}_m)$ be the weight matrix with diagonal entries equal to $\mathbf{w}$. The gradient and Hessian of $\ell(\mathbf{w})$ with respect to $\mathbf{w}$ can be expressed as

$$\nabla \ell(\mathbf{w}) = -\sum_{i=1}^{n} \mathbf{z}_i (y_i - P(\hat{Y} = 1 | Z = \mathbf{z}_i; \mathbf{w})) = \mathbf{Z}^\top (\mathbf{y} - \mathbf{p}), \text{ and}$$

$$\nabla^2 \ell(\mathbf{w}) = -\sum_{i=1}^{n} \mathbf{z}_i \mathbf{z}_i^\top P(\hat{Y} = 1 | Z = \mathbf{z}_i; \mathbf{w})(1 - P(\hat{Y} = 1 | Z = \mathbf{z}_i; \mathbf{w})) = -\mathbf{Z}^\top \mathbf{W} \mathbf{Z}. \tag{SM. 15}$$

We use the Newton–Raphson algorithm to update the weights, i.e.,

$$\begin{aligned} \mathbf{w}^{t+1} &= \mathbf{w}^t - \left(\nabla^2 \ell(\mathbf{w})\right)^{-1} \nabla \ell(\mathbf{w}) \\ &= \mathbf{w}^t + \left(\mathbf{Z}^\top \mathbf{W} \mathbf{Z}\right)^{-1} \mathbf{Z}^\top (\mathbf{y} - \mathbf{p}), \end{aligned} \tag{SM. 16}$$

where $t \in [1, T]$ is the index of the iterations, and $\mathbf{w}^t$ is the weight at iteration t. Note that the features $\mathbf{z}_i$ could be kernel transformation of a sample $\mathbf{x}_i$, logits outputed from a neural network of a sample $\mathbf{x}_i$, or even the sample $\mathbf{x}_i$ itself. When $\mathbf{z}_i = \mathbf{x}_i$, it is the vanilla logistic regression.

In order to perform adversarial weight perturbation on $\mathbf{w}$ (i.e., to maximize scores of different classes in (9)), for a target feature input $\mathbf{z}_t$, when $y_t = 0$, we aim to maximize $\mathbf{w}^\top \mathbf{z}_t$ such that $P(\hat{Y} = 1 | Z = \mathbf{z}_i; \mathbf{w})$ is maximized. Similarly, when $y_t = 1$, we aim to minimize $\mathbf{w}^\top \mathbf{z}_t$ such that $P(\hat{Y} = 0 | Z = \mathbf{z}_i; \mathbf{w})$ is maximized. Therefore, we modify the gradient in (SM. 15) to

$$\nabla \ell(\mathbf{w}) = \mathbf{Z}^\top (\mathbf{y} - \mathbf{p}) + \lambda_t \mathbf{z}_t, \tag{SM. 17}$$

where $\lambda_t$ is a regularization parameter, and $\lambda_t > 0$ if $y_t = 0$, and $\lambda_t < 0$ if $y_t = 1$. When $\lambda = 0$, (SM. 17) degenerates to (SM. 15). Therefore, the adversarial weight perturbation on logistic regression could be performed by keep updating the weights with

$$\mathbf{w}^{t+1} = \mathbf{w}^t + \left(\mathbf{Z}^\top \mathbf{W} \mathbf{Z}\right)^{-1} \left(\mathbf{Z}^\top (\mathbf{y} - \mathbf{p}) + \lambda_t \mathbf{z}_t\right), \tag{SM. 18}$$

until convergence. The reason we introduce the features $\mathbf{z}_i$ in the beginning instead of the samples $\mathbf{z}_i$ if that if that if $\mathbf{z}_i = f(\mathbf{x}_i)$ for a neural network $f(\cdot)$, (SM. 18) can be used for last-layer weight perturbation of the neural network [21].

## SM. 2.7   Convexity of the channel capacity

In the last paragraph of Section 5, we mention an important limitation of KL-divergence based Rashomon Capacity due to the convexity of KL-divergence: in certain cases $C(\mathcal{M}_\epsilon(\mathbf{x}_i))$ (and therefore $m_C(\mathbf{x}_i) = 2^{C(\mathcal{M}_\epsilon(\mathbf{x}_i))}$) may seem small for already significant score variations across the classes. Here, we use an example the binary asymmetric channel [1] to illustrate this phenomenon. Given $p, q \in [0, 1]$, a binary asymmetric channel $X \to Y$ has a channel transition matrix $\mathbf{P} = [[p, 1 - p], [q, 1 - q]] \in [0, 1]^{2 \times 2}$. When $p = q$, the binary asymmetric channel matches the binary symmetric channel. In Fig. SM. 1, we show the channel capacity algorithm, with different pairs $(p, q)$. We observe that the channel capacity is a very "flat" convex function of $p$ and $q$; for example, when $p = 0.5$ and $q = 0.1$, the channel capacity is 1.3, and the channel capacity is larger than 1.8 if the difference $|p - q|$ is larger than 0.7. When the channel transition matrix to be the estimated scores in a binary classification problem, Rashomon Capacity must be interpreted accordingly For example, the difference of the scores of a sample for class 0 and class 1 needs to be larger than 0.7 such that the Rashomon Capacity exceeds 1.8. In fact, a Rashomon capacity above 1.1 already corresponds to a potentially significant score variation in practice.

## SM. 3   Datasets and experiments setups

### SM. 3.1   Dataset descriptions and pre-processing procedures.

**UCI adult dataset.**   The UCI Adult dataset [22] contains multiple domestic factors including an individual's education level, age, gender, occupation, and etc. We drop missing values, and obtain

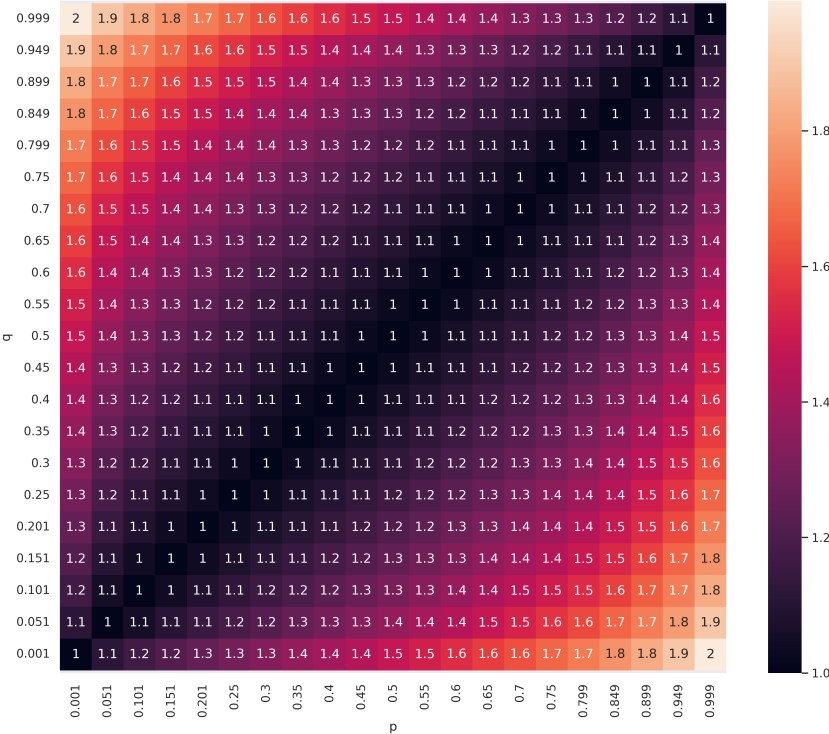

Figure SM. 1: Channel capacity (values annotated on the heatmap) of the binary asymmetric channel with different $p$ and $q$.

46447 samples with 20 features. The 20 features include the one-hot encoded version of the originally selected features [age, education, marital-status, relationship, race, gender, capital-gain, capital-loss, hours-per-week]; note that the features 'race' and 'gender' are binarized. The label is the income, and is divide into two classes: <=50K and >50K.

**COMPAS recidivism dataset.** The COMPAS (Correctional Offender Management Profiling for Alternative Sanctions) dataset [23] is a widely used algorithm for judges and parole officers to score criminal defendant's likelihood of reoffending (i.e., recidivism). The features include [age, charge degree, race, sex, priors crime count, days before screening/arrest, jail in date, jail out date], and the label is the binary prediction on recidivism. We pre-processed the features by binarizing 'race', 'sex', 'charge degree' (felony or others), and 'days before screening/arrest' (<= 30 days or > 30 days); creating a new feature call 'length of stay', which is duration between 'jail in date' and 'jail out date'. The resulting dataset has 52878 samples with 6 features.

**HSLS dataset.** The HSLS (High School Longitudinal Study) dataset [24] is collected from 23,000+ participants across 944 high schools in the USA, and it includes thousands of features such as student demographic information, school information, and students' academic performance across several years. We pre-processed the dataset (e.g., dropping rows with a significant number of missing entries and students taking repeated exams, performing k-NN imputation, normalization), and the number of samples reduced to 14,509 and the number of features is 59. For the labels, we created a binary label Y from students' 9th-grade math test score (i.e., top 50% vs. bottom 50%).

**CIFAR-10 dataset.** The CIFAR-10 dataset [25] contains 50,000 colored images for training and 10,000 for test, where each images has $32 \times 32$ pixels, and has a label 10 classes [airplanes, cars, birds, cats, deer, dogs, frogs, horses, ships, and trucks]. The samples are distributed evenly on the 10 classes for both training and test set.

## SM. 3.2 Training details and experimental setups

For UCI Adult, COMPAS and HSLS datasets, the hypothesis space is composed of simple feed-forward neural networks with ReLU activations, and the optimizer is gradient descent trained with the whole datasets, and the training loss is the cross-entropy loss, and the learning rate is 0.001. For UCI Adult dataset, the neural networks have 5 layers/100 neurons per layer, and is trained with 100 epochs. For COMPAS dataset, the neural networks have 5 layers/200 neurons per layer, and is trained with 200 epochs. For HSLS dataset, the neural networks have 5 layers/200 neurons per layer, and is trained with 500 epochs.

For CIFAR-10 dataset, the hypothesis space is composed of VGG16 convolutional neural networks [26], and the optimizer is stochastic gradient descent with batch size 40. The VGG16 models are trained with the cross-entropy loss for 3 epochs and the learning rate is 0.001.

**Sampling.** For UCI Adult, COMPAS and HSLS datasets, we did 5 repeated experiments with difference random seeds for 70%/30% train/test split, and in each experiments, we trained 100 models, and evaluated on the test set. We select the smallest test loss, and select models that have test losses smaller than the smallest test loss plus the Rashomon parameter $\epsilon = [0.01, 0.02, 0.05, 0.1]$. For CIFAR-10 dataset, we did 2 repeated experiments with difference random seeds for 90%/10% train/test split, and in each experiments, we trained 50 models, and evaluated on the test set. The mean accuracy for UCI Adult, COMPAS, HSLS and CIFAR-10 datasets are $0.8034, 0.6540, 0.6247$ and $0.8380$ respectively. The Rashomon Capacity of all test samples can then be computed by the scores generated by the selected models for difference $\epsilon$, and the mean and standard errors of the largest 1% and 5% Rashomon Capacity, i.e., the statistics on the tails of the Rashomon Capacity, are reported in Fig. 3.

**Adversarial weight perturbation.** For UCI Adult, COMPAS and HSLS datasets, we did 3 repeated experiments with difference random seeds for 95%/5% train/test split, 90%/10% train/test split and 90%/10% train/test split respectively. We first trained a base classifier, and perturbed the weights of the neural networks for each test sample (cf. (9)) with learning rates 0.001 (for UCI Adult and COMPAS datasets) and 0.01 (for HSLS dataset). We require the perturbation procedure to stop updating the weights if either the perturbed scores exceed 0.9, or the test loss is larger than the base test loss plus the Rashomon parameter $\epsilon = [0.01, 0.02, 0.05, 0.1]$. Similarly, for CIFAR-10 dataset, we did 2 repeated experiments with difference random seeds for 99%/1% train/test split. The mean accuracy of the base classifiers for UCI Adult, COMPAS, HSLS and CIFAR-10 datasets are $0.8028$, $0.6458, 0.7039$ and $0.8167$ respectively. Therefore, for each sample, we computed the Rashomon Capacity of all test samples with scores from the base classifier and from the perturbed classifier.

---

**Algorithm SM. 1** Sampling with Rejection

---

**Require:** training set $\mathcal{S}$, test set $\mathcal{T}$, number of models $m \in \mathbb{N}$, Rashomon parameter $\epsilon > 0$
  SampledModel $\leftarrow [\,]$
  TestLoss $\leftarrow [\,]$
  RashomonSet $\leftarrow [\,]$
  RashomonSetProb $\leftarrow [\,]$
  **for** $i \in [m]$ **do**
    model $\leftarrow$ train($\mathcal{S}$, random_seed=i)
    loss $\leftarrow$ evaluate(model, $\mathcal{T}$)
    SampledModel.append(model)
    TestLoss.append(loss)
  **end for**
  **for** $i \in [m]$ **do**
    **if** TestLoss[i] < min(TestLoss) + $\epsilon$ **then**
      RashomonSet.append(SampledModel[i])
      RashomonSetProb.append([compute_scores(SampledModel[i], $\mathcal{T}$)])
    **end if**
  **end for**
  **return** RashomonSet, RashomonSetProb

---

**Algorithm SM. 2** Adversarial Weight Perturbation (AWP)

**Require:** dataset $\mathcal{D} = \{\mathbf{x}_i, \mathbf{y}_i\}_{i=1}^{n}$, pretrained model $f_\theta : \mathcal{X} \to \Delta_c$ with weight $\theta$, learning rate $\gamma$, number of classes $c$

  RashomonSetProb $\leftarrow$ zeros(n, c, c)
  BaseLoss $\leftarrow$ evaluate($f_\theta$, $\mathcal{D}$)
  **for** $i \in [n]$ **do**
    **for** $j \in [c]$ **do**
      CurrentLoss $\leftarrow$ evaluate($f_\theta$, $\mathcal{D}$)
      **while** CurrentLoss < BaseLoss + $\epsilon$ **do**
        scores $\leftarrow f_\theta(\mathbf{x}_i)$
        $\nabla\theta \leftarrow \frac{\partial\text{-scores[j]}}{\partial\theta}$
        $\theta \leftarrow \theta + \gamma\nabla\theta$
        CurrentLoss $\leftarrow$ evaluate($f_\theta$, $\mathcal{D}$)
      **end while**
      RashomonSetProb[i, j, :] $\leftarrow f_\theta(\mathbf{x}_i)$
    **end for**
  **end for**

Table SM. 1: Samples with high Rashomon Capacity in the COMPAS dataset.

| Age | Charge Degree | Race | Sex | Prior Counts | Length of Stay | Max Score | Min Score |
|-----|---------------|------|-----|--------------|----------------|-----------|-----------|
| 25 | 1 | 0 | 1 | 5 | 35 | 0.498 | 0.194 |
| 49 | 0 | 1 | 0 | 2 | 82 | 0.886 | 0.270 |
| 58 | 0 | 1 | 1 | 2 | 83 | 0.885 | 0.224 |
| 45 | 0 | 0 | 1 | 20 | 46 | 0.354 | 0.030 |
| 40 | 0 | 0 | 1 | 24 | 1 | 0.453 | 0.047 |
| 25 | 1 | 1 | 1 | 5 | 101 | 0.756 | 0.196 |
| 45 | 0 | 0 | 0 | 9 | 75 | 0.799 | 0.162 |
| 66 | 1 | 0 | 1 | 33 | 13 | 0.489 | 0.014 |
| 37 | 0 | 0 | 1 | 3 | 80 | 0.867 | 0.301 |
| 58 | 1 | 1 | 1 | 7 | 185 | 0.987 | 0.343 |
| 53 | 1 | 1 | 1 | 9 | 117 | 0.890 | 0.199 |
| 29 | 0 | 0 | 1 | 1 | 99 | 0.930 | 0.416 |
| 37 | 0 | 1 | 1 | 5 | 82 | 0.849 | 0.272 |
| 37 | 0 | 0 | 1 | 22 | 1 | 0.434 | 0.058 |
| 52 | 1 | 0 | 1 | 7 | 117 | 0.921 | 0.251 |

### SM. 3.3    Algorithm boxes

For the sake of clarify, we summarized the sampling and AWP algorithms to explore the Rashomon set and to compute Rashomon Capacity in Algorithm SM. 1 and Algorithm SM. 2 respectively. Both algorithms produce scores from models in the Rashomon set, where the scores are used later to compute the Rashomon Capacity by the Blahut-Arimoto algorithm (Section SM. 2.5).

## SM. 4    Additional experiments

### SM. 4.1    A Case study on the COMPAS dataset

We trained 1k multi-layer perceptron classifiers with different random seeds and selected classifiers which have loss smaller than $0.685$ (the smallest loss observed was $0.68$), and compute Rashomon Capacity. We show the samples with Rashomon Capacity higher than 1.2 in the COMPAS dataset in Table SM. 1. Observe that the samples with conflicting scores are mostly with sex 1 (marked as Male in the dataset) and numerous prior convictions (i.e., prior counts or length of stay). Thus, one must recommend caution when evaluating input samples with this profile. This example showcases how a stakeholder can zoom into samples with high Rashomon Capacity and flag them for further investigation.

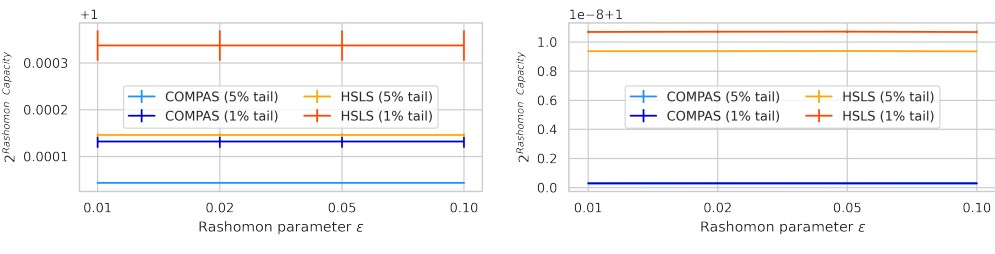

(a) Training with label flipping

(b) Fast Gradient Sign Method (FSGM)

Figure SM. 2: Other methods to explore the Rashomon sets.

## SM. 4.2   Other methods to explore the Rashomon sets

The procedure in (9) reveals a desirable property of a Rashomon subset: it should include models with significant score variations. Similarly, a desirable Rashomon subset for accurately evaluating ambiguity/discrepancy is one with models that have most score disagreement. Based on the observation, we briefly overview next two alternative strategies for identifying a Rashomon subset: training with label flipping [27] and fast gradient sign method (FSGM) [28].

**Training with label flipping**   For training with label flipping, a classifier is trained with a sample whose label is adversarially corrupted to different classes, i.e., training $c$ classifiers for a sample of a $c$-class classification problem, with the goal of producing conflicting scores of the sample. In Fig. SM. 2 (Left), we performed the label flipping procedure and report Rashomon Capacity with different Rashomon parameters $\epsilon$ on 1k random samples in the test set of COMPAS and HSLS datasets. The accuracy of the base classifier and the mean accuracy of the classifiers trained with a flipped label are $0.68029$ and $0.6660$ for COMPAS dataset, and $0.7337$ and $0.7324$ for HSLS dataset. The Rashomon Capacity are all small since a miss classification of a single sample does not significantly influence the overall empirical risk. Therefore, the classifiers are more likely to ignore the sample with a flipped label.

**Fast gradient sign method (FSGM)**   The FSGM is different from the proposed adversarial weight perturbation in (9) in two aspects. First, FSGM is applied to create an imperceivable perturbation on the samples instead of the weights. Second, FSGM only uses the *sign* of the gradient (times a scalar $\beta$) to update the weights. We implemented FSGM on the weights to adversarially change the scores of 1k random samples in the test set of COMPAS and HSLS datasets, and report Rashomon Capacity in Fig. SM. 2 (Right). Note that even with a small scalar $\beta = 0.0001$, the update on the weights—despite being imperceivable when added to input samples—could lead to a significant change of the loss when added to the model weights, and most classifiers updated with the FSGM would not belongs to the Rashomon set defined by the Rashomon parameter. Therefore, Rashomon Capacity is almost 0, as observed in Fig. SM. 2 (Right).

## SM. 4.3   Evaluating Rashomon Capacity in the decision domain

In Figure SM. 3, we demonstrate that Rashomon Capacity can be evaluated with both scores and decisions generated from 100 models in the Rashomon set. Decision-based Rashomon Capacity has integer values, indicating the number of classes that are confused for each sample.

## SM. 4.4   Predictive multiplicity scores based on predicted classes: ambiguity and discrepancy

The computation of the ambiguity and discrepancy in (2) requires searching over the entire Rashomon set, which is computationally infeasible when the hypothesis space is composed of neural networks. However, we can restrict the search in the entire Rashomon set to the sampled Rashomon set, and approximate the ambiguity and discrepancy. In Fig. SM. 4, we report both ambiguity and discrepancy of the 100 sampled models used to produce Fig 3 for UCI Adult, COMPAS, and HSLS datasets. Note that both ambiguity and discrepancy report high predictive multiplicity. For example, in COMPAS dataset, 38% of the samples can be assigned conflicting predictions by switching between classifiers

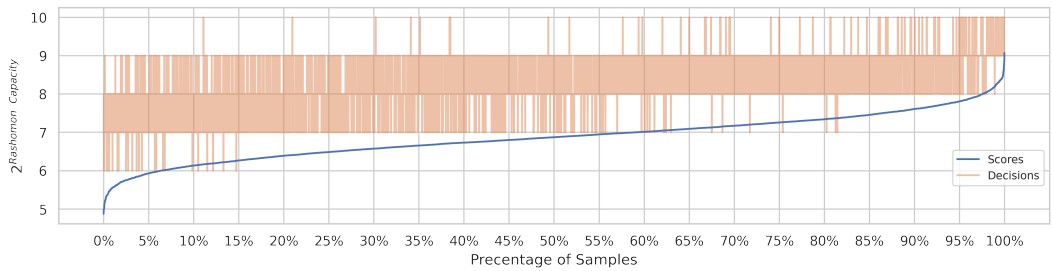

Figure SM. 3: Rashomon Capacity of samples in CIFAR-10 in score and decision domains. The samples are sorted in increasing order of score-based RC. For each sample, we then plot the RC using the corresponding thesholded scores.

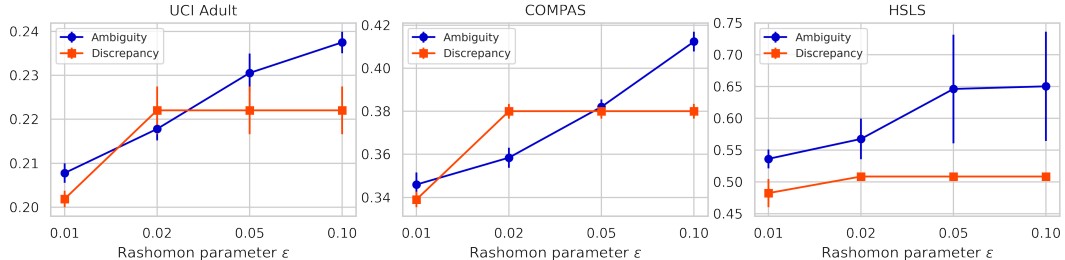

Figure SM. 4: Ambiguity and discrepancy of UCI Adult, COMPAS, and HSLS datasets.

with test loss difference less than 0.05, and in HSLS dataset, the proportion goes to 50%. The reason for such a high predictive multiplicity measured by the ambiguity and discrepancy is that the classifiers, despite having high accuracy, often produce similar scores across the classes for most of the samples.

## SM. 4.5 Rashomon Capacity with ensemble methods and calibrated classifiers

We implement Platt scaling [29] to calibrate the scores produced by classifiers (decision tree and random forest classifiers), and compare the Rashomon Capacity with and without calibration in Table SM. 2. We observe that with calibration, the Rashomon Capacity are slightly reduced when using decision classifiers, and slightly increased when using random forest classifiers (in UCI Adult and HSLS datasets). It indicates that model calibration could be a (partial) solution to reduce multiplicity, and is an interesting future direction.

We would like to further clarify that if a perfectly calibrated classifier assigns a 50% score to a sample (e.g., in binary classification), it does not necessarily mean that this sample has high multiplicity. A perfectly calibrated classifier is one whose predicted classes matches the true classes on average across samples (e.g., samples predicted to be 50% of one class have true outcomes matching that class 50% of the time). However, this does not necessarily translate to a (in)consistent set of predictions for a single target sample across equally calibrated classifiers. It may be the case that all calibrated models drawn from the Rashomon Set assign the same 50% probability for that sample (no multiplicity). Conversely, some models may assign higher and lower confidence for that sample (high multiplicity) yet, on average, still be well-calibrated. Again, this happens because calibration (like accuracy) is an average metric across all samples.

we observe that a random forest classifier leads to a smaller Rashomon Capacity when compared to a decision tree. This is likely due to random forests being an ensemble method. This is in line with two observations: (i) loss functions are often convex, so convex combinations of classifiers will not increase loss, and (ii) score variation (as measured by RC) is captured by at most $c$ models, so a small number of models can reflect multiplicity across the whole Rashomon set. Ensembling is a viable strategy for resolving multiplicity in small models, but may be infeasible for large, computationally expensive models (e.g., neural networks). We will include ensembling as a promising strategy in the Final Remark.

Table SM. 2: Rashomon Capacity with and without model calibration.

| Dataset | Classifier | Rashomon Capacity | | Rashomon Capacity (Cali.) | |
|---|---|---|---|---|---|
| | | 5% tail | 1% tail | 5% tail | 1% tail |
| UCI Adult | Decision Tree | $1.37 \pm 0.16$ | $1.72 \pm 0.02$ | $1.38 \pm 0.16$ | $1.68 \pm 0.01$ |
| | Random Forest | $1.00 \pm 0.00$ | $1.01 \pm 0.00$ | $1.03 \pm 0.03$ | $1.07 \pm 0.02$ |
| COMAS | Decision Tree | $1.04 \pm 0.00$ | $1.05 \pm 0.00$ | $1.00 \pm 0.00$ | $1.00 \pm 0.00$ |
| | Random Forest | $1.00 \pm 0.00$ | $1.00 \pm 0.00$ | $1.00 \pm 0.00$ | $1.00 \pm 0.00$ |
| HSLS | Decision Tree | $1.22 \pm 0.02$ | $1.25 \pm 0.02$ | $1.19 \pm 0.02$ | $1.21 \pm 0.01$ |
| | Random Forest | $1.00 \pm 0.00$ | $1.00 \pm 0.00$ | $1.03 \pm 0.01$ | $1.04 \pm 0.01$ |

## SM. 4.6 Training without neural networks: UCI Adult, COMPAS, and HSLS datasets

We report Rashomon Capacity with learning models that are not neural networks; particularly, we adopt decision tree/ random forest classifiers, and logistic classifiers with no, $\ell_1$, $\ell_2$ and elastic net regularizations, trained with UCI Adult, COMPAS, and HSLS datasets. We sampled 100 classifiers for each model, and report the distribution of Rashomon Capacity among the samples from Fig. SM. 5 to Fig. SM. 19 (UCI Adult: Fig. SM. 5 - Fig. SM. 9; COMPAS: Fig. SM. 10 - Fig. SM. 14; HSLS: Fig. SM. 15 - Fig. SM. 19).

We observe that for all datasets, Rashomon Capacity is significantly reduced with random forest classifiers, comparing to the decision tree classifiers, i.e., predictive multiplicity is alleviated by ensemble methods with a multitude of decision trees methods. On UCI Adult and HSLS datasets, we observe that regularization for logistic regression could also reduce Rashomon Capacity. These preliminary numerical results could serve as future directions on the study of reducing predictive multiplicity via ensemble methods and weight regularization.

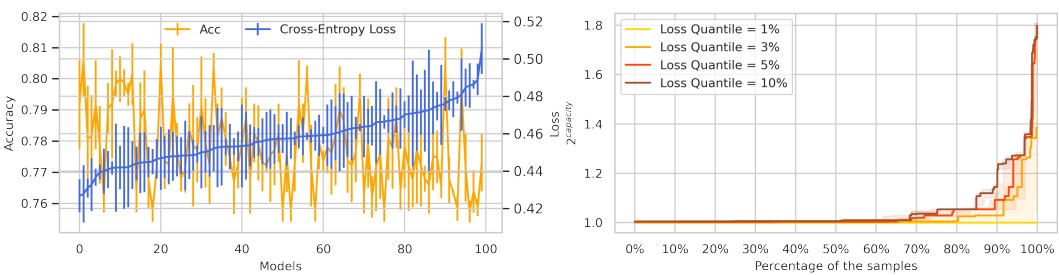

Figure SM. 5: UCI Adult dataset with decision tree classifiers.

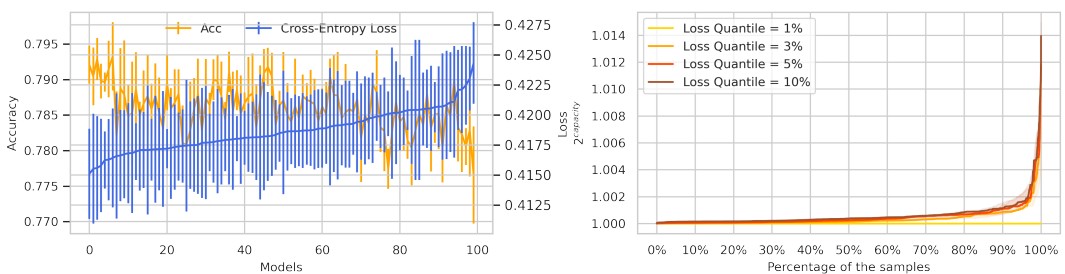

Figure SM. 6: UCI Adult dataset with random forest classifiers.

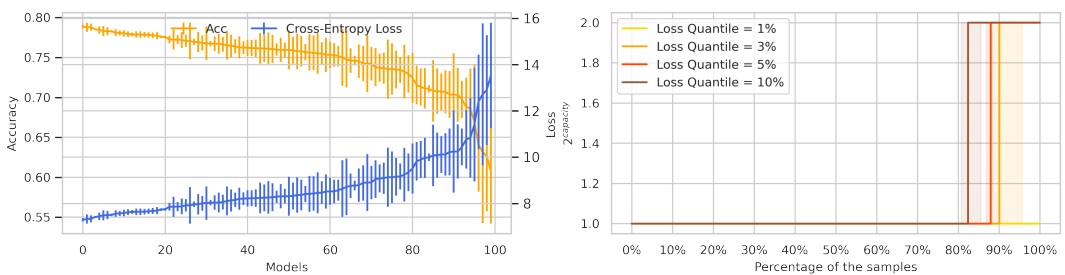

Figure SM. 7: UCI Adult dataset with logistic regression and no regularization.

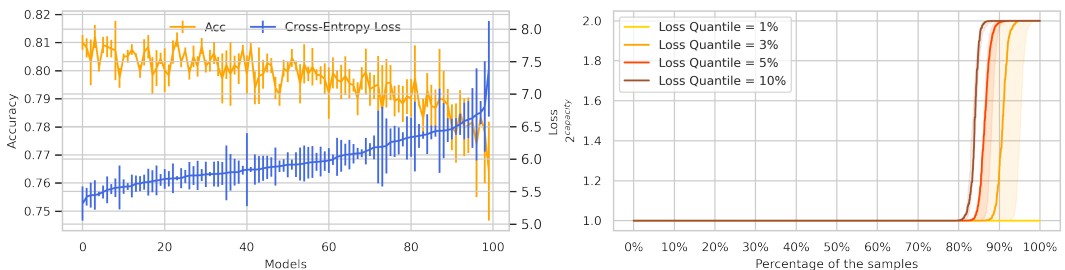

Figure SM. 8: UCI Adult dataset with logistic regression and $\ell_1$-regularization.

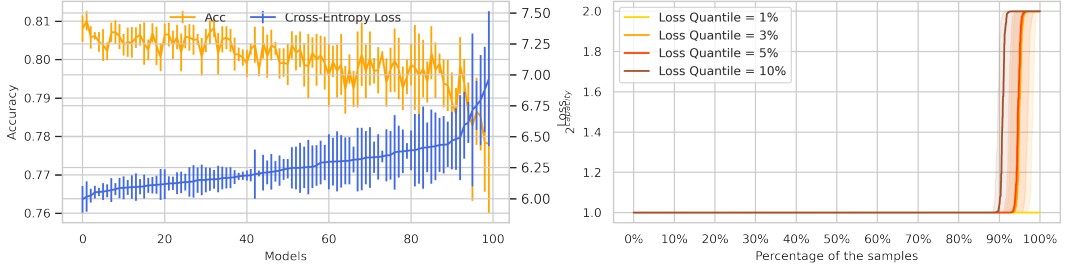

Figure SM. 9: UCI Adult dataset with logistic regression and $\ell_2$-regularization.

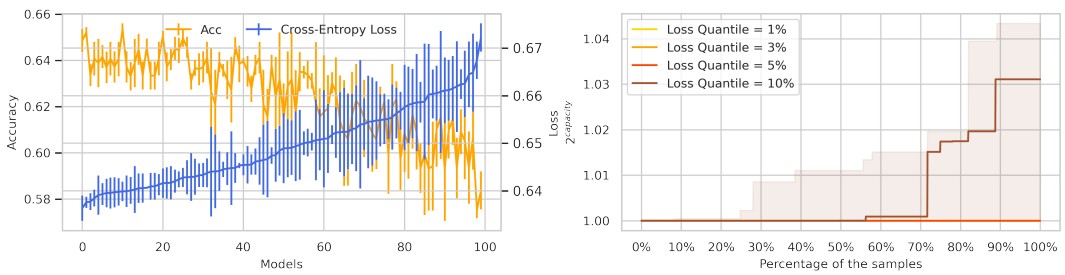

Figure SM. 10: COMPAS recidivism dataset with decision tree classifiers.

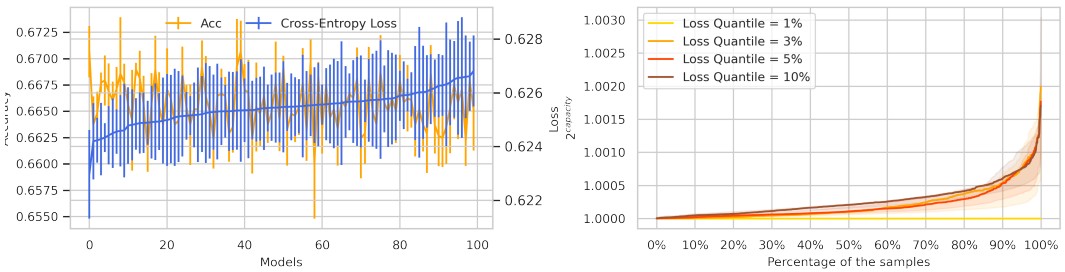

Figure SM. 11: COMPAS recidivism dataset with random forest classifiers.

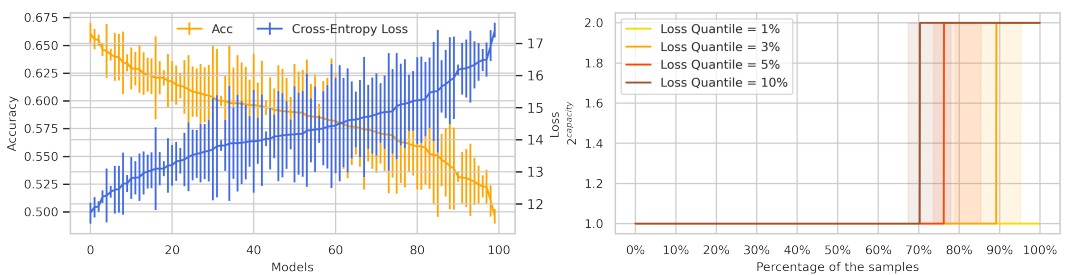

Figure SM. 12: COMPAS recidivism dataset with logistic regression and no regularization.

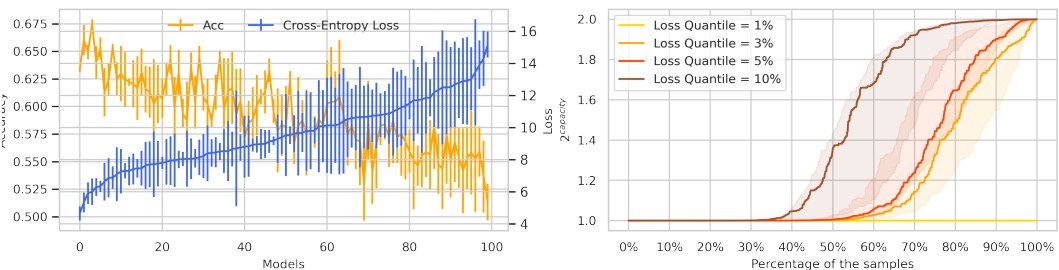

Figure SM. 13: COMPAS recidivism dataset with logistic regression and $\ell_1$-regularization.

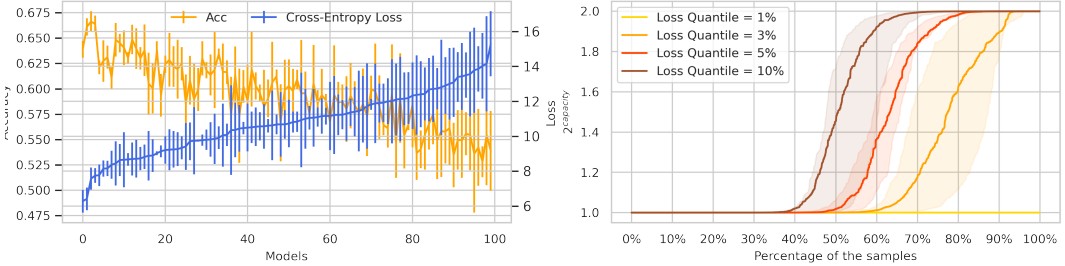

Figure SM. 14: COMPAS recidivism dataset with logistic regression and $\ell_2$-regularization.

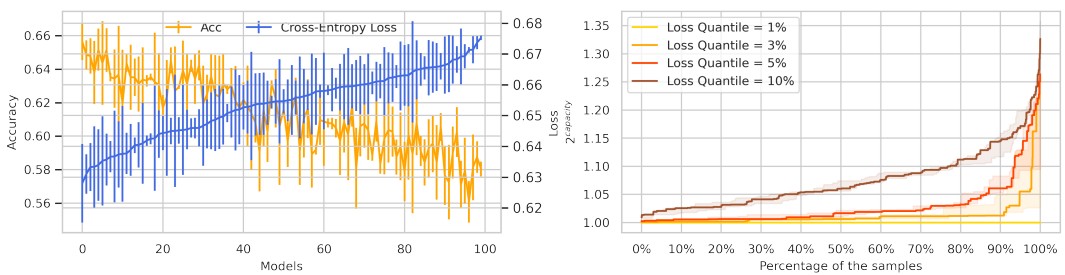

Figure SM. 15: HSLS dataset with decision tree classifiers.

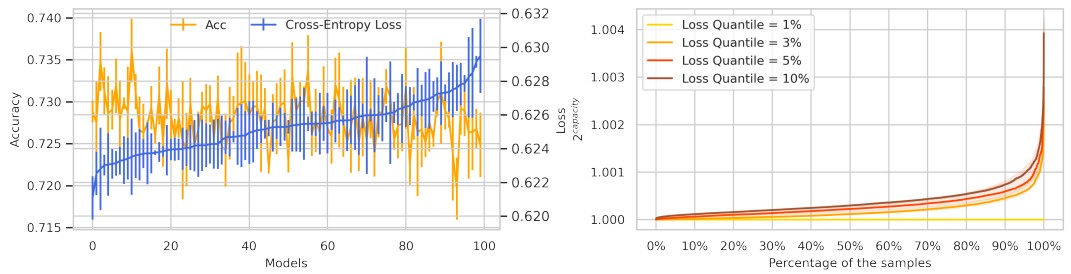

Figure SM. 16: HSLS dataset with random forest classifiers.

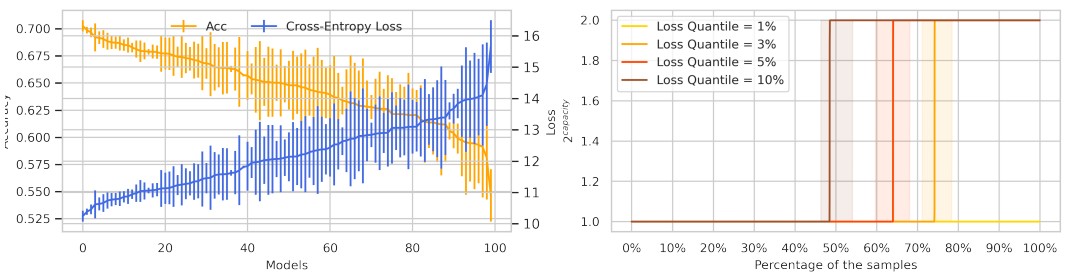

Figure SM. 17: HSLS dataset with logistic regression and no regularization.

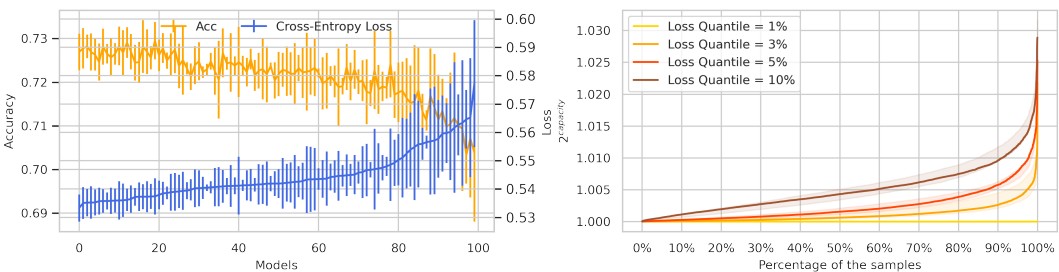

Figure SM. 18: HSLS dataset with logistic regression and $\ell_1$-regularization.

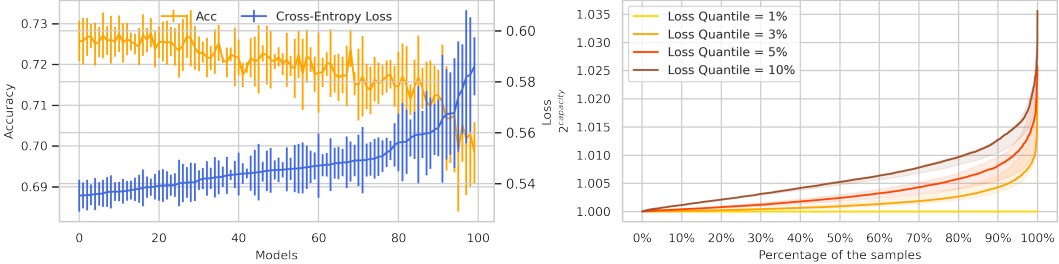

Figure SM. 19: HSLS dataset with logistic regression and $\ell_2$-regularization.