# OpenReview forum: "Rashomon Capacity: A Metric for Predictive Multiplicity in Classification"
_NeurIPS.cc/2022/Conference — NeurIPS 2022 Accept_

### Official Review · Reviewer_1Yaa · 2022-06-27

**Rating:** 5
**Confidence:** 4
**Soundness:** 3 good
**Presentation:** 3 good
**Contribution:** 2 fair

**Summary:**


This paper proposes a new metric to measure predictive multiplicity, called Rashomon Capacity (RC).  Unlike previous metrics, RC depends on the probability prediction instead of the discrete predictions. To compute RC, the authors propose to perform AWP, which is more efficient and random initializations.

**Questions:**

1. How is this paper related to model calibration? If the predicted probability is not calibrated, it seems pointless to use it in the definition/computation of RC.

2. How to choose $\epsilon$ in practice, or how to interpret it? Obviously, the value of RC is critically dependent on $\epsilon$, but it seems like a quantity that's hard to interpret and it's thus not clear to me how to incorporate RC in the decision making process. This is not related to Section 2.3 in the Appendix. (I should mention I'm not generally familiar with how people use Rashomon sets either.)

3. It is not clear whether this paper actually computes a good RC or not. There is a huge gap between AWP and sampling in Fig 3., but does AWP measures RC well or is it still underestimating? Can (and if so how) we tell?


4. Following 1, why avoiding discrete/binary predictions and measuring Rashomon effect in a continuous space is a good thing? Let's say 1000 models all agree a sample $x_i$ has $0.5\pm\epsilon$ probability of being a cat (i.e. a hard example). If we consider the thresholded prediction, then maybe we get high predictive multiplicity due to the high variance around $0.5$. RC will say there is no divergence here really. It seems to me in practice this $x_i$, when used with any of these 1000 models, is still worth a lot of attention from the decision maker, so it should be considered as a high multiplicity input?

**Limitations:**


See Weaknesses above. In general I think the motivation is the major limitation.


Minor issues:
1. typo @L145: "Given" -> "given"

**Strengths And Weaknesses:**


Strengths:

The problem of predictive multiplicity seems important. Proposition 2 makes reporting RC possible in practice. This paper is well-written and easy to follow. The introduction lays a good background of this area.

Weaknesses:

1. RC seems very expensive. Computing RC for one data point requires performing $c$ optimizations. While random initializations can be shared for different data samples and save time, from Fig 3 it is clearly (greatly) underestimating RC.

2. The motivation seems weak, given the cost of computation and the unclear message it conveys without as specific $\epsilon$. It could probably be used to measure model robustness, but I think that's just [31]? Can the authors briefly explain how RC will be used in a decision marking process? My question 4 is also related to motivation.

3. As an evaluation metric, RC (AWP) requires running multiple optimizations. It seems like unlike metrics like accuracy, it is unclear how to compare RC across different $\mathcal{H}$ (or even within the same $\mathcal{H}$ but with different optimization methods) if the optimization is not finding global optima.

---

> ### Author Response · Authors · 2022-08-02
> **Response to Reviewer 1Yaa**
>
> 1. **RC seems very expensive. Computing RC for one data point requires performing c optimizations. While random initializations can be shared for different data samples and save time, from Fig 3 it is clearly (greatly) underestimating RC.**
>
>     We thank the reviewer for raising this point. In summary, the computationally expensive part of estimating RC comes from exploring the Rashomon set. Exploring the Rashomon Set is a  common challenge in all existing works on multiplicity [3, 4]. For neural networks, accurately estimating the Rashomon set amounts to characterizing the number of local minima — a computationally challenging problem in deep learning, since even simple networks trained with quadratic loss have exponentially many minima [R1]. Given an approximation of the Rashomon Set and, more specifically, conflicting outputs produced by models in the Rashomon Set, RC can be efficiently computed by solving a convex optimization problem using the Blahut-Arimoto Algorithm.
>
>     Yes, AWP may still underestimate the true RC, yet it greatly improves estimation compared to prior work, as shown in Figure 3.  AWP outperforms sampling (random initializations) and other methods (label flipping and FGSM): as discussed in SM Section 4.2, both methods underestimate Rashomon Capacity. More importantly, our experiments indicate that AWP less computationally intensive than sampling. A binary classification problem with AWP is guaranteed by Proposition 2 with at most 2 re-trainings per sample. However, the sampling method could require more than 250k re-trainings, as done in [3 Section 6].
>
>     Finally, we also highlight that reporting predictive multiplicity — even if an underestimate — is better than reporting no multiplicity at all (the current practice). As discussed in the introduction and argued by [5], disclosing multiplicity is crucial when models are deployed in sectors that may control access to critical opportunities to large portions of the population (e.g., lending, access to higher education). In such cases, identifying and disclosing predictive multiplicity — even if an underestimate — is certainly better than not disclosing any multiplicity at all (the current practice). In this sense, we believe RC is a step forward in both generality and applicability.
>
>     [R1] Auer, P., Herbster, M. and Warmuth, M.K., 1995. Exponentially many local minima for single neurons. *Advances in neural information processing systems*, *8*.
> **Continued...**

---

> > ### Author Response · Authors · 2022-08-02
> > **Response to Reviewer 1Yaa - Continued**
> >
> > 2. **The motivation seems weak, given the cost of computation and the unclear message it conveys without as specific ϵ. It could probably be used to measure model robustness, but I think that's just [31]? Can the authors briefly explain how RC will be used in a decision marking process? My question 4 is also related to motivation.**
> >
> >     The Rashomon parameter $\epsilon$ captures the statistical uncertainty in estimating loss due to finite samples. In this regard, $\epsilon$ will depend on the number of samples of a test or validation set used to compare models. In our experiments, we chose $\epsilon$ to be the minimal loss achievable with the sampled models plus a small $\epsilon’$ (e.g., 0.05 shown in Figure 3, and cf. Footnote 3). The larger the number of samples in a dataset, the smaller the $\epsilon$ can be made since confidence in model performance increases.
> >
> >     In decision-making processes, we suggest disclosing both score variation and thresholded decisions to stakeholders. We improved the manuscript by including a case study with COMPAS on how to use Rashomon Capacity in a decision-making process in SM Section 4.1.
> >
> >
> > ---
> >
> > 3. **As an evaluation metric, RC (AWP) requires running multiple optimizations. It seems like unlike metrics like accuracy, it is unclear how to compare RC across different H (or even within the same H but with different optimization methods) if the optimization is not finding global optima.**
> >
> >     We thank the reviewer for this interesting question. Indeed, unlike the Rashomon set, which size/volume is directly connected to the complexity of the hypothesis set H (larger H leads to a larger Rashomon set), the value of Rashomon Capacity is not directly related to H. In particular, a more complex H does not necessarily lead to a higher Rashomon Capacity, see Section 3.2 for an example of a larger Rashomon set with a smaller Rashomon Capacity. Ideally, we would like to choose H to yield the most accurate predictions with the least amount of multiplicity.
> >
> >
> > ---
> >
> > 4. **How is this paper related to model calibration? If the predicted probability is not calibrated, it seems pointless to use it in the definition/computation of RC.**
> >
> >     We thank the reviewer for raising this interesting direction. Rashomon Capacity can still be applied to measure the multiplicity of a calibrated classifier. We repeat experiments in Figure 3 with UCI Adult, COMPAS and HSLS dataset with calibrated (Platt scaling) classifiers with decision tree and random forest classifiers, and observe that calibration did slightly reduce Rashomon Capacity, but did not remove predictive multiplicity. See details in Section 4.5 of the revised SM (and table SM.2 in particular). We will add to the main text this discussion on the role of calibration.
> >
> >
> > ---
> >
> > 5. **How to choose ϵ in practice, or how to interpret it? Obviously, the value of RC is critically dependent on ϵ, but it seems like a quantity that's hard to interpret and it's thus not clear to me how to incorporate RC in the decision-making process. This is not related to Section 2.3 in the Appendix. (I should mention I'm not generally familiar with how people use Rashomon sets either.)**
> >
> >     Please see answer to Q2 above and Footnote 3. Here, $\epsilon$ is the loss achieved by, for example, the model that is the empirical risk minimizer or a pre-trained classifier, plus a small factor $\epsilon'$. The value of $\epsilon’$ will depend on the confidence interval for model performance (as measured by a test or validation loss). As mentioned above, the population loss of a classifier can only be approximated by an estimate on a test or validation set. In our experiments, $\epsilon$ is the loss achieved by the best classifier plus a small $\epsilon'$ chosen as constant for simplicity (see Footnote 3). Again, $\epsilon'$ would depend on the size of the test set: when empirical loss is estimated using $n$ samples, usually $\epsilon'$ will be of the order $O(1/\sqrt{n})$. We will add this discussion to Section 3.
> >
> > **Continued...**

---

> > > ### Author Response · Authors · 2022-08-02
> > > **Response to Reviewer 1Yaa - Continued**
> > >
> > > 6. **It is not clear whether this paper actually computes a good RC or not. There is a huge gap between AWP and sampling in Fig 3., but does AWP measures RC well or is it still underestimating? Can (and if so how) we tell?**
> > >
> > >     We thank the reviewer for this great question. The computation of any multiplicity metric, including the ones outlined in the Related Work section, inevitably relies on an efficient exploration of the Rashomon set. As mentioned in Sections 3 and 4, characterization of the Rashomon set is a challenging task — in neural networks, this may amount to characterizing all (exponentially many [R1]) local minima. Therefore, every estimate of Rashomon Capacity based on a subset of the Rashomon set (by either sampling or AWP) will underestimate its value, unless the Rashomon set can be fully characterized.
> > >
> > >     Multiplicity is not resolved unless we are able to find the single model with the smallest population loss in the hypothesis set, which is in general difficult due to a limited number of samples and optimization techniques. Nevertheless, reporting Rashomon Capacity is still critical for models deployed in applications of individual-level consequence (e.g., healthcare, education, lending), even if it is an underestimate! This reporting allows stakeholders to understand if a model output may vary due to arbitrary choices made during training (e.g., random initialization). In this sense, reporting some multiplicity (e.g., using Rashomon Capacity) is better than no reporting at all (the current practice). We improved the manuscript by adding this discussion in the Limitations Section.
> > >
> > >
> > > ---
> > >
> > > 7. **Following 1, why avoiding discrete/binary predictions and measuring Rashomon effect in a continuous space is a good thing? Let's say 1000 models all agree a sample xi has 0.5±ϵ probability of being a cat (i.e. a hard example). If we consider the thresholded prediction, then maybe we get high predictive multiplicity due to the high variance around 0.5. RC will say there is no divergence here really. It seems to me in practice this xi, when used with any of these 1000 models, is still worth a lot of attention from the decision maker, so it should be considered as a high multiplicity input?**
> > >
> > >     We agree with the reviewer’s comment. In fact, Rashomon Capacity can be computed for **both** scores and thresholded decisions! This follows from the fact that one-hot encoded model outputs still lie on the probability simplex. Thus, given conflicting (thresholded) outputs, RC can be computed using the exact procedure in the paper. We will clarify this point in Section 3 and add reporting of RC for both scores and thresholded outputs in the SM.
> > >
> > >     We would like to emphasize that we advocate that multiplicity should be measured at **both** the score level and threshold level. In fact, as we have mentioned in footnote 6, both metrics should be reported as complementary metrics for characterizing multiplicity and paint different pictures on how predictive multiplicity occurs. Taking ternary classification as an example, for three score vectors [0.49, 0.51,0] and [0.51, 0.49,0], our suggestion is to report Rashomon Capacity for both thresholded **and** the non-thresholded scores. In this case, the multiplicity (given by Rashomon Capacity) of the raw scores for these samples is close to 1. For the thresholded scores ([0,1,0] and [1,0,0]) the Rashomon Capacity is now 2. Note that this is also revealing: since RC is between 1 and 3, this indicates that the confusion is between 2 classes instead of 3. In contrast, prior metrics that also operate on thresholded scores, such as ambiguity and discrepancy (Eq. 2) do not yield a less nuanced picture, capturing only if predictions agree or not. Rashomon Capacity is capable of measuring the different spread of scores. In this sense, score-level metrics could potentially provide a **finer** characterization of predictive multiplicity.
> > >
> > >     We have added additional comparisons with other multiplicity metrics in Section 4.4 of the SM.
> > >
> > >
> > > Thanks again for the review! We would be happy to provide more clarifications and answer any follow-up questions.

---

> > ### Comment · Reviewer_1Yaa · 2022-08-07
> > **Additional Comments**
> >
> >
> >
> > I thank the authors for the response. While I'm still digesting some responses, I would like to post some of my new comments first.
> >
> > ### 4
> >
> > Thank you for adding the experiment for calibration.
> > I did not mean that calibration will remove multiplicity (although it is interesting to see the effect), but I just meant you could probably discuss the relation in the paper.
> > In my opinion, if a perfectly calibrated classifier says an image is 50\% likely to be a cat, this value itself (instead of the variance of it) says something about (high) multiplicity.
> >
> > ### 3 and 6
> > I appreciate additional discussion on RC's relation to $\mathcal{H}$ and the difficulty of estimating RC.
> > I understood what the authors explained, but my major concern here is actually that RC seems not able to **fairly** compare different classifiers.
> > AWP is optimization based, so the quality seems to naturally depend on $\mathcal{H}$.
> > (In general I feel the difficulty of exploring the Rashomon Set (RS) depends on $\mathcal{H}$.)
> > What if the RS for DNNs are just harder to exlore, so we systematically underestimate its RC?
> > Do we know if this is or is not the case?
> > "Accuracy", for example, is model independent, so we do not need to worry about these issues and could just use it as a "metric" for classifiers.
> >
> > Maybe a more fundamental question is:
> > Is RC supposed to be used to compare different **samples** under the same class of classifier (e.g. two images in CIFAR-100 but with the same ResNet-50) or compare different **classifiers** (e.g. DNN vs Random Forest)?
> > From the title of Section 3, and the paper as a whole, I thought it's the latter, but I still don't see RC-AWP could/should be used as a metric to compare across different classifiers basing on the above.
> >
> > ### 7
> > Thank you pointing out footnote 6. Could you point me to where you emphasized that **both** should be used in Section 3 as well? I think this is not very clear from the main text (as I think reviewer 9SyE has a similar question).

---

> > > ### Author Response · Authors · 2022-08-08
> > > **Response to Reviewer 1Yaa’s Additional Comments**
> > >
> > > (**4) Thank you for adding the experiment for calibration. I did not mean that calibration will remove multiplicity (although it is interesting to see the effect), but I just meant you could probably discuss the relation in the paper. In my opinion, if a perfectly calibrated classifier says an image is 50% likely to be a cat, this value itself (instead of the variance of it) says something about (high) multiplicity.**
> > >
> > > We thank the reviewer again for the suggestion on adding experiments for model calibration, and will add the discussions on the effect of model calibration in the main text!
> > >
> > > We would like to further clarify that if a perfectly calibrated classifier assigns a 50% score to a sample (e.g., in binary classification), it does not necessarily mean that this sample has high multiplicity. A perfectly calibrated classifier is one whose predicted classes matches the true classes **on average** across samples (e.g., samples predicted to be 50% of one class have true outcomes matching that class 50% of the time). However, this does not necessarily translate to a (in)consistent set of predictions for a **single target sample** across equally calibrated classifiers. It may be the case that **all** calibrated models drawn from the Rashomon Set assign the same 50% probability for that sample (no multiplicity). Conversely, some models may assign higher and lower confidence for that sample (high multiplicity) yet, on average, still be well-calibrated. Again, this happens because calibration (like accuracy) is an average metric across all samples.
> > >
> > > In summary, we plan to add the following points to the final version of the paper:
> > >
> > > 1) Pointer to the experiment on how calibration impacts multiplicity (already in the SM);
> > >
> > > 2) A brief discussion, matching the one above, on how calibration — being an average metric like accuracy — does not necessarily resolve multiplicity, though may reduce it as we observe in the new experiments.
> > >
> > > 3) The importance of disclosing scores, thresholded scores, and multiplicity. This is particularly relevant in applications such as medicine, where thresholded scores are translated into medical decisions and raw scores into medical risk assessments. Here, multiplicity can shed light if the model output is a fluke due to arbitrary choices made during training or is consistent across models in the chosen hypothesis space.
> > >
> > > **Continued...**

---

> > > > ### Author Response · Authors · 2022-08-08
> > > > **Response to Reviewer 1Yaa’s Additional Comments - Continued**
> > > >
> > > > **(3 and 6) I appreciate additional discussion on RC's relation to $\mathcal{H}$ and the difficulty of estimating RC.I understood what the authors explained, but my major concern here is actually that RC seems not able to fairly compare different classifiers.AWP is optimization based, so the quality seems to naturally depend on $\mathcal{H}$.(In general I feel the difficulty of exploring the Rashomon Set (RS) depends on $\mathcal{H}$.)What if the RS for DNNs are just harder to exlore, so we systematically underestimate its RC?Do we know if this is or is not the case?"Accuracy", for example, is model independent, so we do not need to worry about these issues and could just use it as a "metric" for classifiers. Maybe a more fundamental question is: Is RC supposed to be used to compare different samples under the same class of classifier (e.g. two images in CIFAR-100 but with the same ResNet-50) or compare different classifiers (e.g. DNN vs Random Forest)?From the title of Section 3, and the paper as a whole, I thought it's the latter, but I still don't see RC-AWP could/should be used as a metric to compare across different classifiers basing on the above.**
> > > >
> > > > This is an excellent point! We thank the reviewer for bringing up this issue, and would like to clarify the confusion here. In Section 3, we assume that the Rashomon set and the hypothesis space (i.e., a model class) are fixed, and propose the definition of Rashomon Capacity of a sample given that hypothesis space. We did not intend to use Rashomon Capacity for comparing among different hypothesis spaces and purposefully chose not to compare multiplicity across different model classes in this work. In fact, we use Rashomon Capacity to compare the multiplicity of **different samples** in a dataset for the **same** model class. The 4 methods (sampling [3], AWP, label flipping and FGSM) we used to explore the Rashomon set in Section 4 are all performed on the same hypothesis space, e.g., the same neural network architecture. We would like to emphasize that the full characterization of the Rashomon set is generally infeasible (even for a very simple hypothesis space, e.g., decision trees), and thus all estimates of multiplicity metrics so far are underestimates. Moreover, we empirically showed that the AWP algorithm outperforms existing methods in terms of estimating multiplicity metrics.
> > > >
> > > > The reason we purposefully chose not to report results that compare across model classes is because we share the same concern as the reviewer: Though our work is the first to provide methods for measuring and reporting multiplicity in complex models (e.g., neural networks), and we leap beyond prior work which mostly focused on linear models/decision trees, we are **not** yet at a stage where we can responsibly say that certain model classes consistently yield "lower" multiplicity and may thus be preferable in practice. We will add this warning to the final section of the paper.
> > > >
> > > > Indeed, if one cannot systematically characterize the Rashomon Set across different hypothesis spaces (e.g., via the AWP algorithm), then the Rashomon Capacity of a sample should not be used to compare different hypothesis spaces. However, this does **not** mean that multiplicity should not be measured for a given model class. The goal here is to ensure that outputs of a model communicated to stakeholders (e.g., patients) are consistent for a chosen hypothesis space, and not due to an arbitrary choice made during training.
> > > >
> > > > We note that similar issues appear even when comparing **accuracy** across classifiers, since the same hypothesis space (e.g., a given neural network architecture) could potentially yield higher accuracy with different methods of searching that space (e.g., via better optimization techniques). In fact, the problem of further optimizing accuracy for a fixed hypothesis space is a long lasting challenge in machine learning research.
> > > >
> > > > We will add a summary of the discussion above to the limitations section of the paper, and thank the reviewer for pressing this point.
> > > >
> > > > **Continued...**

---

> > > > > ### Author Response · Authors · 2022-08-08
> > > > > **Response to Reviewer 1Yaa’s Additional Comments - Continued**
> > > > >
> > > > > **(7) Thank you pointing out footnote 6. Could you point me to where you emphasized that both should be used in Section 3 as well? I think this is not very clear from the main text (as I think reviewer 9SyE has a similar question).**
> > > > >
> > > > > We recognize we could have done a better job emphasizing that Rashomon Capacity can be applied to both thresholded and raw scores in the main text, and note that this is a simple fix since it does not change any of our code/mathematical formulations. We will highlight this point in the main text — currently it is stated in the SM section 4.3. To fix this, we will move footnote 6 to the main text, and emphasize that Rashomon Capacity can be computed for **both** scores and thresholded decisions, and that multiplicity metrics on score and threshold decisions are both important. More specifically, we shall emphasize this in Section 3, prior to the start of Section 3.1, and also in Section 3.3 when describing how to compute Rashomon Capacity. Please also see responses to ****Reviewer hCae (3)**** for more comparisons between Rashomon Capacity and existing metrics.
> > > > >
> > > > > We hope that we have addressed all your concerns! Please let us know if you have further questions, and we really appreciate your input in making our submission a better and clearer paper.

---

> ### Author Response · Authors · 2022-08-05
> **Thanks again for your review and time!**
>
> Thanks again for your thoughtful review! We believe that we have addressed all of your concerns and questions in our response above. We would love to receive any additional feedback you may have. Do you have any follow-up questions? We are excited to engage in further discussions this week! Please let us know.
> Thank you very much, and we look forward to hearing from you!

---

### Official Review · Reviewer_9SyE · 2022-07-06

**Rating:** 4
**Confidence:** 4
**Soundness:** 3 good
**Presentation:** 3 good
**Contribution:** 2 fair

**Summary:**

The paper proposes ways to quantify “predictive multiplicity” (the extent to which equally accurate models disagree on individual predictions) for probabilistic classifiers. Specifically, it introduces a quantity called Rashomon Capacity and demonstrates its usage to report predictive multiplicity on different datasets and models.


**Questions:**

- I would like to see the authors strengthen the claim that we need to model predictive multiplicity at the score level (as opposed to the decision level).
- Some questions/comments regarding the definition of the Rashomon set: first, I would distinguish the true Rashomon set (computed w.r.t the true risk) and the empirical Rashomon set (computed w.r.t the empirical risk). E.g. I would denote them $R(H,\epsilon)$ and $\hat{R}(H, \epsilon)$, respectively. More importantly, I don’t understand why the $\epsilon$-level set is one sided - i.e., consists of all the classifiers in $H$ that have loss at most $\epsilon$ as opposed to loss approximately $\epsilon$ . I understand that for a specific choice of $\epsilon$  (close to the loss of the ERM, as mentioned in the footnote) the two notions coincide, but (a) on a definitional level, it’s not clear to me why this is the definition, (b) in the experiments, a range of values of $\epsilon$  is considered. I find this weird because if the best loss achievable in $H$ is 0.1 (e.g. by $h1$) then the Rashomon set w.r.t $\epsilon = 0.2$ will naturally will have different predictions for some inputs (when $h1$ disagrees with the predictions of $h2$, a classifier with loss $0.2$), but it’s not clear to me that this falls under the “predictive multiplicity” phenomena (because $h1$ and $h2$ are clearly distinguishable).
- In the definition of desirable properties for a metric $m(\cdot)$, is it important that there is a classifier that assigns the one-hot vector $e_j$ as opposed to that the argmax is $j$?


**Limitations:**

Yes

**Strengths And Weaknesses:**

Strengths:
- The problem of predictive multiplicity is important and timely.
- The paper is written quite clearly.
- The notion of Rashomon capacity is natural & well motivated.

Weakenesses:

- In terms of significance, I found the main argument for extending existing notions for binary classifiers to probabilistic classifiers not completely clear/satisfying. For example, it is mentioned that two scores can be very close (e.g. one example has score 0.49 for class 1, the other 0.51), and the existing methods would report multiplicity when there isn’t “really” any multiplicity. But doesn’t this depend on how the probabilistic classifier ends up being used? E.g. if the probabilistic classifier is the extracted probabilities from a Logistic regression model that will then be used to make binary decisions by thresholding the scores at 0.5, then I *should* treat 0.49 different from 0.51, no? I think this approach would make sense if binary decisions are obtained by randomizing according to the scores (e.g. classifying someone as 1 w.p 0.49), but I don’t think anyone uses ML models this way.
- In terms of clarity: (1) The paper makes some bold claims which I felt were unjustified, since the claims themselves were not really made formal. For example, it is mentioned in several places that the proposed methodology “resolves” predictive multiplicity. But what does “resolve” even mean in this context? (2) In many places I felt like important details were omitted from the main text. For example, it is mentioned somewhat in passing, “Rashomon Capacity can be computed by standard procedures such as the Blahut-Arimoto algorithm”. The details of the algorithms are fine to omit, but a description of the assumptions is important - for example, it’s not clear to me if this assumes the Rashomon Set itself is already known, and what’s the computational complexity of this procedure. Similarly for the following result (Proposition 2): the statement claims the existence of at most c models, but can they also be found efficiently?
- Finally, in terms of quality: The connection between the definitions and theoretical results to the experimental section was often unclear or lacking. For example, if I understand correctly, the proposed methods (e.g. sampling and AWP) are heuristics, and do not compute “the Rashomon capacity” but some approximation of it. I couldn’t really understand if sampling is a baseline and AWP is the “proposed” approach and whether the claim is that it is better (because the capacities are larger). Also I would have liked to see some semi-synthetic experiments (where the “real” Rashomon capacity is known or can be computed efficiently) so that the performance of the heuristics could be evaluated. E.g., for all we know, the Rashomon capacity is large for every input and we only observe it’s small on some inputs because of optimization issues.

Overall I think the paper could have high impact, but at its current form I'm left with more questions than answers regarding many of the concepts used and the claims made, hence my current score.

---

> ### Author Response · Authors · 2022-08-02
> **Response to Reviewer 9SyE**
>
> We thank the reviewer for the feedback. We clarify the questions point-by-point below.
>
> 1. **In terms of significance, I found the main argument for extending existing notions for binary classifiers to probabilistic classifiers not completely clear/satisfying. For example, it is mentioned that two scores can be very close (e.g. one example has score 0.49 for class 1, the other 0.51), and the existing methods would report multiplicity when there isn’t “really” any multiplicity. But doesn’t this depend on how the probabilistic classifier ends up being used? E.g. if the probabilistic classifier is the extracted probabilities from a Logistic regression model that will then be used to make binary decisions by thresholding the scores at 0.5, then I *should* treat 0.49 different from 0.51, no? I think this approach would make sense if binary decisions are obtained by randomizing according to the scores (e.g. classifying someone as 1 w.p 0.49), but I don’t think anyone uses ML models this way.**
>
>     We thank the reviewer for raising this important point. We would first like to highlight that Rashomon Capacity can be applied to measure multiplicity in both the score domain and decision domain (i.e., post-thresholding)! This follows from the fact that a one-hot encoded classifier output (e.g., 0 or 1) still lies on the probability simplex, and thus the formulation in Eq. 6 is still valid. This holds independently of how scores are converted to decisions (e.g., thresholding, argmax, etc.). To illustrate this point, we have added examples of RC with thresholded scores in Fig. SM. 3.
>
>     We also note that scores are frequently used when machine learning models are deployed for risk prediction in, for example, medical applications (e.g.,  triaging) and criminal justice (e.g., the COMPAS recidivism prediction instrument would output a (quantized) risk score instead of a binary output).
>
>     Moreover, the proposed methods for exploring the Rashomon Set (e.g., AWP) can also be applied given that the raw scores are thresholded after perturbation. We will make this point clearer in the revised version of the manuscript in Section 4.
>
>     We do not advocate for only reporting multiplicity in the score domain, as stated in Footnote 6. Taking ternary classification as an example, for three score vectors [0.49, 0.51,0] and [0.51, 0.49,0], our suggestion is to report Rashomon Capacity for both thresholded **and** the non-thresholded scores. In this case, the multiplicity (given by Rashomon Capacity) of the raw scores for these samples is close to 1. For the thresholded scores ([0,1,0] and [1,0,0]) the Rashomon Capacity is now 2. Note that this is also revealing: since RC is between 1 and 3, this indicates that the confusion is between 2 classes instead of 3. In contrast, prior metrics that also operate on thresholded scores, such as ambiguity and discrepancy (Eq. 2) do not yield a less nuanced picture, capturing only if predictions agree or not. Rashomon Capacity is capable of measuring the different spread of scores. In this sense, score-level metrics could potentially provide a **finer** characterization of predictive multiplicity. We have added additional comparisons with other multiplicity metrics in Section 4.4 of the revised SM.
>
>     In summary, the Rashomon Capacity metric in Eq. 6 allows for outputs that are 0 or 1 for each class and the computation does not change! We will clarify this in Section 3 of the revised paper that our methods can be applied to **both** thresholded and non-thresholded scores. We also added an example on computing Rashomon Capacity using thresholded decisions in the SM.
>
> **Continued...**

---

> > ### Author Response · Authors · 2022-08-02
> > **Response to Reviewer 9SyE - Continued**
> >
> > 2. **In terms of clarity:**
> >     - **The paper makes some bold claims which I felt were unjustified, since the claims themselves were not really made formal. For example, it is mentioned in several places that the proposed methodology “resolves” predictive multiplicity. But what does “resolve” even mean in this context?**
> >
> >         We thank the reviewer for bringing up this point.
> >
> >         We use the term "resolving" in the sense of "resolving a dispute" (i.e., deciding a course of action) and “separate or cause to be separated into components" (a sense often used in Physics/Chemistry). Resolving predictive multiplicity requires answering the question “*given many competing models in the Rashomon Set, which prediction(s) should I select for a target sample?”* Predictive multiplicity can be resolved by, for example, selecting a subset of models (potentially of size one) and displaying the scores to a user, or combining the output of competing classifiers via randomization or bagging.
> >
> >         Rashomon Capacity provides a way of selecting a small subset of models which capture the variation of conflicting scores. For each sample, this subset's size is at most the size of the number of predicted classes (as proved in Prop. 2) *regardless of the size of the Rashomon Set*. This small number of output scores can be communicated to a stakeholder, thus helping them decide how to resolve predictive multiplicity.
> >
> >         For example, in Fig. 4a, given 163 models taken from the Rashomon set, one way of resolving multiplicity is to randomly select one output from the 163 models, as suggested in [5], or use ensemble methods (e.g., bagging, see Section 1). Instead of generating one model, we propose a new strategy: report a subset of models that capture the most score variation for a sample. Figure 4(a) shows that instead of reporting/ensembling 163 models, or randomly selecting one model, we only need to select 10 models that capture the majority of score variations across all samples. These 10 models can be released to stakeholders for selection, combination, or to identify samples that have high Rashomon Capacity, thus helping resolve multiplicity in the aforementioned sense.
> >
> >         Please note that, in the revised manuscript, we included an additional worked-out example of estimating Rashomon Capacity in the COMPAS dataset in SM Section 4.1.
> >
> >  **Continued...**

---

> > > ### Author Response · Authors · 2022-08-02
> > > **Response to Reviewer 9SyE - Continued**
> > >
> > > 2. **In terms of clarity:**
> > >     - **In many places I felt like important details were omitted from the main text. For example, it is mentioned somewhat in passing, “Rashomon Capacity can be computed by standard procedures such as the Blahut-Arimoto algorithm”. The details of the algorithms are fine to omit, but a description of the assumptions is important - for example, it’s not clear to me if this assumes the Rashomon Set itself is already known, and what’s the computational complexity of this procedure. Similarly for the following result (Proposition 2): the statement claims the existence of at most c models, but can they also be found efficiently?**
> > >
> > >         We thank the reviewer for pointing this out! We improved the manuscript by adding algorithm boxes for AWP and for computing Rashomon Capacity (including details of the Blahut-Arimoto algorithm) in the revised SM. We also will make the following clarifications in the paper:
> > >
> > >         - There are two separate tasks in computing predictive multiplicity: (i) estimating/approximating the Rashomon Set, and (ii) given this approximation, reporting score variation. Task (i) is a core challenge across all methods that deal with predictive multiplicity and a computationally intensive task. Prior work estimates the Rashmomon set by restricting the model class to linear classifiers and solving constrained integer programs (Marx et al.) or sampling from the Rashomon Set via retraining with random initializations (Semenova et al.). In contrast, our AWP method can be applied to any differentiable model and does not require retraining. Despite being a step forward in applicability from prior work, it is still computationally intensive. We will clarify this point in Section 4.
> > >         - Given an approximation of the Rashomon Set in (i), the next step is to compute predictive multiplicity. Unless the Rashomon Set is fully characterized — a seemingly infeasible task for large models — any metric for multiplicity will be an **underestimate.** But this does not mean that reporting and resolving multiplicity is not important! Even if underestimated, recognizing and disclosing variation in scores is critical when models are deployed in applications of individual-level consequences. As described in the introduction, the alternative is to risk arbitrary and unjustified decisions towards stakeholders. We will add this point to the limitations section.
> > >         - The claim in Proposition 2 is that, given a set of conflicting models for predicting $c$ classes, at most $c$ models will capture the score variation for a sample. The set of conflicting models may be only an approximation of the Rashomon Set, yet the proposition still holds. Naturally, when the set of conflicting models is a strict subset of the Rashomon Set, yielding an underestimate of multiplicity. We will a brief remark after the proposition to clarify this point.
> > >
> > > **Continued...**

---

> > > > ### Author Response · Authors · 2022-08-02
> > > > **Response to Reviewer 9SyE - Continued**
> > > >
> > > > 3. **Finally, in terms of quality: The connection between the definitions and theoretical results to the experimental section was often unclear or lacking. For example, if I understand correctly, the proposed methods (e.g. sampling and AWP) are heuristics, and do not compute “the Rashomon capacity” but some approximation of it. I couldn’t really understand if sampling is a baseline and AWP is the “proposed” approach and whether the claim is that it is better (because the capacities are larger). Also I would have liked to see some semi-synthetic experiments (where the “real” Rashomon capacity is known or can be computed efficiently) so that the performance of the heuristics could be evaluated. E.g., for all we know, the Rashomon capacity is large for every input and we only observe it’s small on some inputs because of optimization issues.**
> > > >
> > > >     In the revised version, we will a clearer distinction on when the Rashomon Set is approximated and yields an underestimation of the true Rashomon Capacity in Section 4. As mentioned in the answer to the next question, the "real" Rashomon Capacity can only be approximated for large and over-parametrized model classes (e.g., neural networks).
> > > >
> > > >     We would like to emphasize that approximating the Rashomon set is a core challenge across the estimation of any multiplicity metric. In this work, we discussed four methods to explore the Rashomon set to find models with conflicting scores/ predictions, including (1) training with label flipping, (2) Fast Gradient Sign Method (FGSM), (3) sampling, and (4) AWP. Particularly, for label flipping, we flipped the label of a sample to the wrong labels and trained it (e.g., for a sample with label 1, we flipped it to 0, 2, 3, etc.), and for FGSM, we update the weights with the sign of the gradient obtained from the optimization problem in Eq. 8. The results of both (1) and (2) were included in SM Section 4.2. AWP outperforms the rest of the methods in terms of discovering models with conflicting scores. Moreover, Proposition 2 provided that, in theory, we only need to perform AWP $c$ times for a single sample.
> > > >
> > > >     We again reiterate that, even though Rashomon Capacity underestimates the true variation in scores due to an approximation of the Rashomon Set, it is still critical to disclose its value in applications of individual-level consequence. As argued by [5], disclosing multiplicity is crucial when models are deployed in sectors that may limit access to critical opportunities to large portions of the population (e.g., lending, access to higher education). In such cases, identifying and disclosing predictive multiplicity — even if an underestimate — is certainly better than the current practice of not disclosing multiplicity at all.
> > > >
> > > > **Continued...**

---

> > > > > ### Author Response · Authors · 2022-08-08
> > > > > **Response to Reviewer 9SyE’s Questions**
> > > > >
> > > > > We add below further clarifications on the reviewer's questions. Thanks again for your review and effort! We would be more than glad to address further comments.
> > > > >
> > > > > **(1) I would like to see the authors strengthen the claim that we need to model predictive multiplicity at the score level (as opposed to the decision level).**
> > > > >
> > > > > We thank the reviewer for this great suggestion! We would like to re-emphasize that Rashomon Capacity can operate on both score and decision levels, since Rashomon Capacity can be computed with any set of elements on the probability simplex, and a decision (i.e., thresholded score) is a “vertex” in the probability simplex. Moreover, scores and decisions paint different pictures when studying predictive multiplicity. As we have explained in the previous response for Q1, score-level metrics could potentially provide a **finer** characterization of predictive multiplicity. In contrast, prior metrics that operate on **aggregate** thresholded scores, such as ambiguity and discrepancy (Eq. 2) render a less nuanced metric, capturing only if predictions across classifiers agree or not. Finally, using score level metrics allows more flexible optimization strategies for characterizing the Rashomon set, since score level metrics are easier to optimize over compared to decisions (e.g., 0-1 loss is non-differentiable).
> > > > >
> > > > > Nevertheless, we highlight that one should not restrict computing multiplicity to score or decision level — it is important to report both. We will emphasize that Rashomon Capacity can be applied to both thresholded and raw scores in the main text (currently it is only mentioned in the SM Section 4.3), and note that this is a straightforward update since it does not change any of our code/mathematical formulations. Specifically, we will move Footnote 6 to the main text,  emphasize that Rashomon Capacity can be computed for **both** scores and thresholded decisions in Section 3, and highlight in the final section that multiplicity metrics on score and threshold decisions should both be reported (e.g., on model cards).
> > > > >
> > > > > **Continued...**

---

> > > > > > ### Author Response · Authors · 2022-08-08
> > > > > > **Response to Reviewer 9SyE’s Questions - Continued**
> > > > > >
> > > > > > **(2) Some questions/comments regarding the definition of the Rashomon set: first, I would distinguish the true Rashomon set (computed w.r.t the true risk) and the empirical Rashomon set (computed w.r.t the empirical risk). E.g. I would denote them $\mathcal{R}(\mathcal{H}, \epsilon)$ and $\hat{\mathcal{R}}(\mathcal{H}, \epsilon)$, respectively. More importantly, I don’t understand why the $\epsilon$-level set is one sided - i.e., consists of all the classifiers in $\mathcal{H}$ that have loss at most $\epsilon$ as opposed to loss approximately $\epsilon$. I understand that for a specific choice of  $\epsilon$ (close to the loss of the ERM, as mentioned in the footnote) the two notions coincide, but (a) on a definitional level, it’s not clear to me why this is the definition, (b) in the experiments, a range of values of  is considered. I find this weird because if the best loss achievable in $\mathcal{H}$ is 0.1 (e.g. by $h_1$) then the Rashomon set w.r.t $\epsilon = 0.2$  will naturally will have different predictions for some inputs (when $h_1$ disagrees with the predictions of $h_2$, a classifier with loss 0.2), but it’s not clear to me that this falls under the “predictive multiplicity” phenomena (because $h_1$ and $h_2$ are clearly distinguishable).**
> > > > > >
> > > > > > We thank the reviewer for the suggestion! We will make the difference between the empirical Rashomon set and population (true) Rashomon set clear, and apply different notations for them in the revision. Note that throughout the paper, we focus on the empirical Rashomon set since the true Rashomon set can only be approximated. The four methods (sampling [3], AWP, label flipping and FGSM) introduced in Section 4, are all algorithms to estimate/approximate the true Rashomon set.  Specifically, we will update the definition of Rashomon Set in Section 2 to stress the empirical version suggested by the reviewer.
> > > > > >
> > > > > > The reason why we adopted a one-sided definition for the Rashomon set is the following. If one were able to find the single, best classifier that outperforms all others in terms of population loss or accuracy, then you are justified to use the classifier that is more accurate in practice. In fact, this point is eloquently made in [5], which argues that there is no (moral) harm in using the most accurate classifier, i.e., selecting a classifier that has a provable smaller loss (or higher accuracy) than another. Thus, it is hard to justify a comparison between classifiers with loss $L_1\pm \delta$ and $L_2 \pm \delta$ for small $\delta$ and loss $L_1\ll L_2$ — since $L_2$ has worse performance, it would not be selected in practice. Please also note that existing works on multiplicity, e.g., [3] and [4], also use the one-sided Rashomon set definition for the same reason.
> > > > > >
> > > > > > Predictive multiplicity arrises when you have multiple classifiers with small and **statistically** indistinguishable losses that are near the minimal loss (see Footnote 3).  In our experiments, the (empirical) Rashomon set are classifiers that are statistically indistinguishable (in loss/accuracy) from a “best” classifier, where the best classifier is the one that achieves the minimum empirical loss. Here, the estimated empirical loss over a dataset is only accurate up to a confidence interval (see SM Figs SM.6-20 for bootstrapped intervals), thus rendering model performance statistically indistinguishable across the Rashomon Set. Thus, $\epsilon$ will depend on this confidence interval, though in our experiments we vary its values to illustrate the impact on the Rashomon Set. **Continued...**

---

> > > > > > > ### Author Response · Authors · 2022-08-08
> > > > > > > **Response to Reviewer 9SyE’s Questions - Continued**
> > > > > > >
> > > > > > > **(continued from the previous post...)** In the experiment (Figure 3), we showcase the average top 1% and top 5% of the Rashomon Capacity from all samples for different $\epsilon$. The reason we chose different values of $\epsilon$ was to showcase how the RC metric varies with this parameter. For example, for UCI Adult dataset, $\epsilon = 0.01$ does not lead to a very different result when $\epsilon = 0.02$, but for COMPAS dataset, the difference in Rashomon Capacity is bigger. In our experiments, we sampled 100 models with different random initializations from $\mathcal{H}$, and determined if a model belongs to the Rashomon set $\mathcal{R}(\mathcal{H}, \epsilon)$ with a pre-set $\epsilon$. For example, when $\epsilon = 0.01$, the models in the Rashomon set $\mathcal{R}(\mathcal{H}, 0.01)$ are indistinguishable since there empirical risk are small and statistically indistinguishable from each other; however, the Rashomon Capacity is non-zero for a significant fraction of samples. Again, in practice, this value should be chosen based on a confidence interval around the minimum, illustrated in the figures in Section 4.7 of the SM. For example, in Figure SM.7, note that the (bootstrapped) confidence intervals for both loss and accuracy **overlap** across models sampled in the Rashomon Set, thus rendering them statistically indistinguishable. We will emphasize this point in Section 2 and 3 of our paper, and note that this is a simple extension of material already covered in the SM.
> > > > > > >
> > > > > > > In the reviewer’s example, for two models $h_1$ and $h_2$ in the Rashomon set $\mathcal{R}(\mathcal{H}, 0.2)$, their losses could be close to each other (e.g., $|L(h_1) - L(h_2)| \sim 0$), or very distant (e.g., $|L(h_1) - L(h_2)| \sim 0.2$). However, in this case, if $\epsilon$ captures the finite-sample uncertainty in estimating loss, both models achieve statistically indistinguishable performance. In practice, both models are equally justified for deployment, since we cannot confidently say which one is, on average, better in terms of accuracy or loss. Again, in practice, $\epsilon$ may be chosen to be small so as to measure the multiplicity of “the set of most accurate models”. If more samples are available to evaluate model performance (rendering more confident performance estimates), this parameter's value will decrease.
> > > > > > >
> > > > > > > We will add this discussion on the choice of $\epsilon$ and the definition on the Rashomon set to the revision and clarify the confusing points.
> > > > > > >
> > > > > > > **(3) In the definition of desirable properties for a metric $m(\cdot)$, is it important that there is a classifier that assigns the one-hot vector $e_j$ as opposed to that the argmax is $j$?**
> > > > > > >
> > > > > > > Yes, the properties are defined on the singleton sets in the probability simplex $\Delta_c$. Note that there is **no** loss in generality in this definition: in practice, one can trivially convert the predicted labels, i.e., the argmax of score vectors, back to one-hot vectors and compute Rashomon Capacity on the one-hot vectors. For example, if decisions are made by taking the argmax and the resulting class is $j$, then the one-hot vector is $e_j$, i.e., the $j$-th standard basis in $\mathbb{R}^c$. In this sense, Definition 1 is more general, since it is independent on **how** the one-hot vectors are produced (e.g., it could be done via different thresholds for different classes or using the argmax). Note that existing metrics on predicted labels, i.e., ambiguity and discrepancy, do not satisfy the desired properties in Definition 1. Please also see responses to ****Reviewer hCae (3)**** for more comparisons between Rashomon Capacity and existing multiplicity metrics evaluated with thresholded outputs.
> > > > > > >
> > > > > > > Thanks again! We are happy to provide further clarifications, and really appreciate your thoughtful input.

---

> ### Author Response · Authors · 2022-08-05
> **Thanks again for your review and time!**
>
> Thanks again for your thoughtful review! We believe that we have addressed all of your concerns and questions in our response above. We would love to receive any additional feedback you may have. Do you have any follow-up questions? We are excited to engage in further discussions this week! Please let us know.
> Thank you very much, and we look forward to hearing from you!

---

### Official Review · Reviewer_hCae · 2022-07-11

**Rating:** 7
**Confidence:** 3
**Soundness:** 3 good
**Presentation:** 4 excellent
**Contribution:** 3 good

**Summary:**

The paper is focused on predictive multiplicity. Predictive multiplicity is when all the models in the Rashomon set (set of models with similar performance statistics on the test set) have different predictions for the same input. The goal is to define a new metric (Rashomon Capacity) for predictive multiplicity that considers the soft probabilistic outputs of the classifier models rather than the one-hot class predictions. This helps to distinguish Rashomon sets where different predictions are a result of highly different predicted probabilities vs  sets where the predictions are different but the come from similar soft outputs. The general idea is to find a prior distribution over members of the Rashomon set such that the divergence from the "center of gravity" of predictions is maximize. The paper continues by computing and approximating the Rashomon capacity metric on various real-world datasets using a several algorithms.

**Questions:**

- What is the theoretical guarantee for the upper bound on errors of the two AWP and sampling algorithms?
- One approach to address the issue seems to be using an ensemble of models. How does the cardinality of Rashomon set scale vs the number of models in the ensembles?
- Another approach to address the issue is to use several metrics rather than cross-entropy loss (fairness metrics, etc). How does the cardinality of Rashomon set change with this approach?

**Limitations:**

* The main algorithm the results in the finding the largest Rashomon Capacity (AWP) is computationally expensive.

------After rebuttal-----

I thank the authors for providing a comprehensive rebuttal. I will take their response to my review and other reviews into consideration throughout the decision process.

**Strengths And Weaknesses:**

* The target problem of the paper is important and well-motivated.
* The paper is well-organized and the writing is smooth. The paper would be easier to understand with a step-by-step algorithm box either in the main text rather than referring to the algorithm's name and the provided code.
* The experimental section focuses on useful data set examples but no case study has been discussed to have a better understanding of how Rashomon capacity can be crucial in real-world scenarios. It would be very useful to have an example.
* The paper seems to be an extension of existing work. In order to prove its contribution, there needs to be an either empirical or theoretical comparison with existing work. in the experimental section no comparative analysis is performed to empirically showcase the advantage of using Rashomon capacity versus ambiguity and discrepancy.
* It would be more comprehensive if the greedy algorithm's performance is compared for the other two datasets (CIFAR10 and adult income).
* All in all, the paper does a decent job in motivating and explaining the Rashomon Capacity but does not provide enough empirical evidence on its advantage and utility in order to satisfy the paper's motivation.

---

> ### Author Response · Authors · 2022-08-02
> **Response to Reviewer hCae**
>
> We thank the reviewer for the feedback and encouragement. We address the questions point-by-point below.
>
> 1. **The paper is well-organized and the writing is smooth. The paper would be easier to understand with a step-by-step algorithm box either in the main text rather than referring to the algorithm's name and the provided code.**
>
>     We thank the reviewer for this suggestion and have added algorithm boxes in the SM Section 3.3 of the revised submission and will point the readers to the algorithm in Section 3.3 of the main text. The algorithms that we added are how to explore the Rashomon set to find conflicting scores for a sample by (1) sampling the Rashomon Set via random initializations, and (2) AWP. The steps for the BA algorithm are presented in SM 2.4.
>
> ---
>
> 2. **The experimental section focuses on useful data set examples but no case study has been discussed to have a better understanding of how Rashomon Capacity can be crucial in real-world scenarios. It would be very useful to have an example.**
>
>     Thank you for the excellent suggestion. We have added to SM 4.1 a worked-out example based on the COMPAS dataset on how Rashomon Capacity can be used to identify high-multiplicity samples. Here, we observe patterns in sex and prior convictions leading to samples having high Rashomon Capacity.
>
>
> ---
> **Continued...**

---

> > ### Author Response · Authors · 2022-08-02
> > **Response to Reviewer hCae - Continued**
> >
> > 3. **The paper seems to be an extension of existing work. In order to prove its contribution, there needs to be an either empirical or theoretical comparison with existing work. in the experimental section no comparative analysis is performed to empirically showcase the advantage of using Rashomon capacity versus ambiguity and discrepancy.**
> >
> >     We thank the reviewer for bringing up this point and will add a clearer description of the comparison with existing work both conceptually and empirically in Section 3 and 4, including the worked-out example mentioned above. We elucidate next the main differences (and potential advantages) of Rashomon Capacity in terms of the most closely related prior work, namely Semenova et al. [3] and Marx et al. [4]. The discussion below complements the details in related literature section of the paper and in the SM.
> >
> > - (Pattern) Rashomon Ratio [3]:
> >     - **Conceptual comparison:** Rashomon ratio and pattern Rashomon ratio are defined in terms of volumes of the hypothesis (or model parameter) space. In contrast, Rashomon Capacity aims to measure the multiplicity of classifier outputs (be it scores or decisions) for individual samples. We measure the spread of the scores with divergence measures for probability distributions. This distinction has been clarified in Section 3.
> >     - **Empirical Comparison:** For models of medium to large size (e.g., neural networks), the Rashomon ratio will be close to zero (e.g, there are more than 100m parameters in VGG16, leading to an extremely large hypothesis space). The evaluation of pattern Rashomon ratio also involves the computation with 2^{number of samples} terms, e.g., for 100 samples, it requires the computation of 2^{100} different combinations for binary classification. In contrast, Rashomon Capacity can be tractably computed by solving a convex optimization problem given outputs from models in the Rashomon Set. Moreover, even though the number of models in the Rashomon Set may be large, Proposition 2 guarantees that at most $c$ models capture the score variation, where $c$ is the number of predicted classes.
> > - Ambiguity/Discrepancy [4]
> >     - **Conceptual comparison:** Ambiguity and discrepancy capture multiplicity in decisions, i.e.,  thresholded scores.  As mentioned in main text Section 2,  ambiguity measures the proportion of samples in a dataset that can be assigned conflicting predictions by
> >     competing classifiers in the Rashomon set. The discrepancy, in turn, is the maximum number of predictions that could change in a dataset if we were to switch between models within the Rashomon set. Like ambiguity and discrepancy, Rashomon Capacity can also be evaluated with decisions, since a one-hot encoded classifier outputs still lie in the probability simplex. In fact, ambiguity measures the fraction of samples in a dataset with non-zero Rashomon Capacity. Rashomon Capacity provides a more nuanced view of score variation, and can be used to identify the number of "conflicting" classes in the predictions produced by models in the Rashomon Set (i.e., RC can be a number between $1$ and $c$). Ambiguity and discrepancy do **not** satisfy the properties of a predictive multiplicity metric outlined in Definition 1.
> >     - **Empirical comparison**: Despite ambiguity and discrepancy being defined for general classifiers, [3] only demonstrates their computation with linear classifiers. It is unclear how to extend the linearly-constrained integer programs in [3, Section 3] to other model classes such as a neural network. For the sake of completeness, we report the ambiguity and discrepancy computed with a sampled Rashomon set with UCI Adult, COMPAS and HSLS datasets in SM Figure 4. Observe that these experiments show that both ambiguity and discrepancy report high predictive multiplicity. For example, in COMPAS dataset, 38% of the samples can be assigned conflicting predictions by switching between classifiers with test loss difference less than 0.05, and in HSLS dataset, the proportion goes to 50%.
> >
> > ---
> >
> > 4. **It would be more comprehensive if the greedy algorithm's performance is compared for the other two datasets (CIFAR10 and adult income).**
> >
> >     We have added the performance of the greedy policy for UCI Adult and CIFAR-10 datasets in the SM (Fig. SM. 5). This figure shows that the greedy policy can also capture the distributions of Rashomon Capacity with fewer models on UCI Adult and CIFAR-10 datasets.
> >
> >
> > **Continued...**

---

> > > ### Author Response · Authors · 2022-08-02
> > > **Response to Reviewer hCae - Continued**
> > >
> > > 5. **All in all, the paper does a decent job in motivating and explaining the Rashomon Capacity but does not provide enough empirical evidence on its advantage and utility in order to satisfy the paper's motivation.**
> > >
> > >     We thank the reviewer for recognizing our efforts in explaining the Rashomon effect and motivating this research problem. As discussed in the answer to Question 2, we clarify the difference between measuring multiplicity on scores and on decisions (i.e., thresholded scores), and that Rashomon Capacity can also be evaluated on 0-1 classifier outputs. We also provide a case study on samples with high Rashomon Capacity using the COMPAS dataset in Table SM.1. Also, please note the additional experiments in the SM Section 4.3, which include the evaluation of Rashomon Capacity with (thresholded) decisions on CIFAR-10. We would be happy to provide further comparisons if requested!
> > >
> > >
> > > ---
> > >
> > > 6. **The main algorithm the results in the finding the largest Rashomon Capacity (AWP) is computationally expensive.**
> > >
> > >     The expensive computation of Rashomon Capacity comes from the exploration of the Rashomon set, which is a core and common problem in all existing methods for characterizing multiplicity. In practice, the Rashomon set (as defined in Eq. 1) can only be approximated. Compared to other methodologies to explore the Rashomon set, for instance, sampling [3], training with label flipping and the Fast Gradient Sign Method evaluated in SM (Section 4.2), AWP is capable of finding models with most score variations in the Rashomon set. Note that, given a set of conflicting models, computing Rashomon Capacity is very tractable and amounts to solving sequences of convex optimization problems via the Blahut-Arimoto algorithm. We will add this discussion in the limitations section.
> > >
> > > **Continued...**

---

> > > > ### Author Response · Authors · 2022-08-07
> > > > **Response to Reviewer hCae - Continued**
> > > >
> > > > 7. **What is the theoretical guarantee for the upper bound on errors of the two AWP and sampling algorithms?**
> > > >
> > > >     Thank you for raising this point! We are unaware of any prior work or method for bounding the error in approximating Rashomon sets. For example, the sampling algorithm to explore the Rashomon set, proposed in [3, Section 5.3], does not carry known theoretical guarantees in terms of the number of samples needed to accurately estimate its value. When the hypothesis space is large, a huge number of sampled models are required to approximate the Rashomon set; for example in [3], 250k decision trees with depth 7 are sampled and the corresponding Rashomon ratio is around 10^{-60}. This is common practice in prior work. The exception is when the model class is convex and constraints that encode "flipping" a decision on an individual sample can be added to during re-training (see Marx et al. [3] for an example in linear classifiers]).
> > > >
> > > >     For neural networks, accurately estimating the Rashomon set amounts to characterizing the number of local minima — a computationally challenging problem in deep learning, since even simple networks trained with quadratic loss have exponentially many minima [R1]. Here, a precise characterization of the Rashomon set seems out of reach.
> > > >
> > > >     Even though a theoretical guarantee remains elusive across existing works, we note that we take an important step towards understanding and approximating predictive multiplicity: Proposition 2 implies that, in theory, at most $c$ models capture all score variation for a sample (as measured by Rashomon Capacity). This theoretical result is the motivation for our AWP method, which requires re-training at most $c$ models. Contrast this with prior approaches: Marx et al. is restricted to linear classifiers and requires solving sequences of integer programs for each point in the dataset. We agree that our method is a heuristic, but it is inspired by theory and, unlike previous efforts, sheds light on how multiplicity can be estimated in complex models (e.g., neural networks). To the best of our knowledge, methods for estimating predictive multiplicity in large model classes have not been proposed outside of simple retraining with random initializations — which we benchmark against in Figure 3.
> > > >
> > > >     We would be happy to discuss further, and reiterate again that, even though Rashomon Capacity underestimates the true variation in scores due to an approximation of the Rashomon set, it is still critical to disclose its value in applications of individual-level consequence.
> > > >
> > > >     [R1] Auer, P., Herbster, M. and Warmuth, M.K., 1995. Exponentially many local minima for single neurons. *Advances in neural information processing systems*, *8*.
> > > >
> > > >
> > > > ---
> > > >
> > > > 8. **One approach to address the issue seems to be using an ensemble of models. How does the cardinality of Rashomon set scale vs the number of models in the ensembles?**
> > > >
> > > >     This is a fascinating and important question. In the SM (Table SM.2 and Section 4.6), we observe that a random forest classifier leads to a smaller Rashomon Capacity when compared to a decision tree. This is likely due to random forests being an ensemble method. This is in line with two observations: (i) loss functions are often convex, so convex combinations of classifiers will not increase loss, and (ii) score variation (as measured by RC) is captured by at most $c$ models, so a small number of models can reflect multiplicity across the whole Rashomon set. Ensembling is a viable strategy for resolving multiplicity in small models, but may be infeasible for large, computationally expensive models (e.g., neural networks). We will include ensembling as a promising strategy in the Final Remark.
> > > >
> > > > ---
> > > >
> > > > 9. **Another approach to address the issue is to use several metrics rather than cross-entropy loss (fairness metrics, etc). How does the cardinality of Rashomon set to change with this approach?**
> > > >
> > > >     The definition of the Rashomon set has only one constraint on the loss (Eq. 1), and if we apply more constraints (e.g., fairness metrics [18]) to define the Rashomon set, the size of the Rashomon set will decrease. However, considering that the hypothesis space is very large (e.g., more than 100m parameters in VGG16), the resulting Rashomon set with an additional fairness constraint could still be very large and computationally challenging to characterize. We also highlight the important recent work [18], which uses a reductions approach to produce an accurate model in the "fair Rashomon Set.” We will add this discussion to the Related Work section of our paper.
> > > >
> > > > Thanks again for the constructive review! We would be excited to provide more clarifications and answer any follow-up questions you may have.

---

> ### Author Response · Authors · 2022-08-05
> **Thanks again for your review and time!**
>
> Thanks again for your thoughtful review! We believe that we have addressed all of your concerns and questions in our response above. We would love to receive any additional feedback you may have. Do you have any follow-up questions? We are excited to engage in further discussions this week! Please let us know. Thank you very much, and we look forward to hearing from you!

---

### Author Response · Authors · 2022-08-02
**Thank you for your constructive feedback!**

We would like to thank the reviewers for their time and effort in reading and commenting on the manuscript. We appreciate that the reviewers found the paper "well-motivated and organized" (Reviewer hCae), the “theoretical results make reporting Rashomon Capacity possible in practice" (Reviewer 1Yaa), and that the problem is "important and timely" (Reviewer 9SyE). We also appreciate their questions, comments, and suggestions. We have addressed all of the reviewers' major and minor comments (including typos), and outlined changes that we have addressed in the **revised version of the supplementary material (SM)**. The changes to the original text are highlighted in blue in the pdfs of the SM.

Please feel free to follow up! We very much welcome further discussions.

---

### Author Response · Authors · 2022-08-09
**Thank you again for your time and feedback!**

Thank you once more for your comments and time. We're writing since we are now nearing the end of the author-reviewer discussion period.
If you have additional comments, please let us know soon and we would be happy to try to answer before the deadline. If we have addressed your questions, we would appreciate it if you would kindly consider raising your score.
Thanks again!

---

### Meta-Review · Area_Chair_eu6x · 2022-08-25

**Recommendation:** Accept
**Confidence:** Less certain

**Metareview:**

The paper presents a new metric for “predictive multiplicity”, which is the tendency of different models from a hypothesis class with similar overall performance to make different predictions on individual samples; predictive multiplicity is relevant to fairness and interpretability of ML models. The paper also presents analysis and algorithms for computing the metric.

The three reviewers generally agreed in their characterization of the paper: the high-level goal was well motivated and timely, the paper was very well written, and technically solid. They raised concerns/questions about motivation and connection to other ideas (e.g., ensembles, calibration), as well as specific suggestions for the experiments and writing. The authors made substantive changes in response to suggestions (especially those of Reviewer hCae) and wrote very detailed responses to questions. Overall, the remaining hesitation from reviewers centers on the significance and usefulness of these ideas in practice. This left the paper as borderline in its ratings. To the meta-reviewer (who also looked at the paper), some skepticism about whether this is the final solution for characterizing the reliability of ML model predictions is certainly warranted; however, the paper appears to be a solid contribution to a nascent area that provides a starting point and is likely to provoke discussion and follow-on work.


**Award:**

No

---

### Decision · Program_Chairs · 2022-09-14

Accept